# Rethinking the Hardness of PbRL: A Provable General Regret Bound

**Chenjie Mao**[1]   **Yi Fan**[2]   **Ning Zhang**[1]   **Chongjie Zhang**[1]

## Abstract

This paper studies *preference-based reinforcement learning* (PbRL), where agents learn from comparative, trajectory-level feedback rather than numeric rewards. While PbRL has seen rapid empirical and theoretical progress, existing analyses are largely confined to restricted settings and fail to jointly capture the outcome-based and comparison-based nature of preference feedback. We prove that under a broad *general function approximation* framework, PbRL admits a $\sqrt{T}$ regret guarantee. In particular, we introduce a simple and provably efficient algorithm, *Recursive Trajectory-based Preference Q-Learning* (RTPQ), and establish its regret bound while explicitly accounting for the trajectory-level and comparative structure of preferences. Our analysis is characterized by a new complexity measure, the *Dual Episodic Eluder Dimension* (DEED), which quantifies the intrinsic difficulty of PbRL. We show that for linear MDPs, the DEED scales as $\mathcal{O}(dH)$, yielding a regret bound of $\widetilde{\mathcal{O}}(dH\sqrt{T}\max\{H^{3/2}, 1/\kappa\})$, where $\kappa$ is a problem-dependent constant. This bound is near-optimal up to horizon- and problem-dependent factors when compared to standard reward-based linear MDPs. In addition, our framework recovers the best-known regret bounds in the special cases of dueling bandits and standard outcome-based reinforcement learning. Overall, our results provide a general regret guarantee for PbRL with outcome-based preference feedback and broad function approximation.

## 1. Introduction

Preference-based reinforcement learning (PbRL) has emerged as a powerful paradigm in modern machine learning. A prominent example is the post-training of large language models (Ouyang et al., 2022; Bai et al., 2022; Rafailov et al., 2024; OpenAI et al., 2024), where human preferences are collected at the end of an interaction by comparing two generated outputs. This approach has gained traction because it replaces the need for a carefully designed reward function with preference comparisons that are easier and more practical to obtain, thereby substantially reducing the cost and complexity of reward engineering.

However, the theoretical understanding of PbRL remains limited. On one hand, existing analyses often rely on restrictive structural or knowledge assumptions, such as tabular MDPs (Novoseller et al., 2020; Xu et al., 2020; Pacchiano et al., 2023; Shani et al., 2024; Zhang et al., 2024; 2025), linear representations (Zhan et al., 2024; Wu & Sun, 2024; Du et al., 2024; Schlaginhaufen et al., 2025), Bayesian formulations (Wu & Sun, 2024; Qi et al., 2025), or other specialized structures (Wang et al., 2023; Xie et al., 2024; Chen et al., 2025b; Zhang & Ying, 2025). On the other hand, methods in the general setting require complex algorithms that iterate across all possible horizons with distinct policies (Xu et al., 2020; Chen et al., 2025a), while some suffer from unfavorable sample complexity (Novoseller et al., 2020; Chen et al., 2022). Overall, despite progress, a general and scalable theoretical framework for PbRL under realistic assumptions remains elusive.

Given these challenges inherent to preference feedback, it is natural to ask:

*Can we develop an algorithm that is general to structural settings, leverages the intrinsic structure of PbRL, and enjoys sample efficiency?*

This paper answers this question affirmatively by proposing a *value-based* algorithm, termed *Recursive Trajectory-based Preference Q-Learning (RTPQ)*, which directly optimizes the value function using trajectory-level preference signals. Our algorithm incorporates the following key designs:

- **Trajectory-wise value estimation:** The value function is estimated at the trajectory level, based on the inferred return function from preference data.

- **Comparison-based updates:** The estimation process explicitly captures the comparative nature of preference feedback.

---

[1]Washington University in Saint Louis [2]Amazon (the work does not relate to their position at Amazon). Correspondence to: Chongjie Zhang <chongjie@wustl.edu>.

*Proceedings of the 43rd International Conference on Machine Learning*, Seoul, South Korea. PMLR 306, 2026. Copyright 2026 by the author(s).

- **Value-centric learning:** The algorithm focuses on estimating and exploring the value function directly, thereby avoiding explicit model learning and planning.

- **Trajectory-level exploration:** Exploration (optimism) is conducted at the trajectory level, ensuring that potential estimation errors of the value function are propagated across the entire trajectory.

- **Consecutive trajectory comparison:** The algorithm compares consecutive trajectories, promoting stable improvement and enhancing robustness.

Under the general function approximation setting, we show that our algorithm achieves a $\sqrt{T}$ regret bound. Our regret analysis introduces a new notion, the *dual episodic eluder dimension (DEED; Definition 2.11)*, which measures the expressiveness and generalization capacity of the value function class within the underlying problem. To further concretize our result, we demonstrate that in the classic linear MDP setting (see Jin et al. (2020)), the DEED scales as $\widetilde{\mathcal{O}}(dH)$ where $d$ is the feature dimension and $H$ is the horizon. Consequently, the regret simplifies to $\widetilde{\mathcal{O}}(dH\sqrt{T}\max\{H^{3/2}, 1/\kappa\})$ where $\kappa$ is a problem dependent constant, which is only a factor of $\widetilde{\mathcal{O}}(\max\{H, 1/(\kappa\sqrt{H})\})$ worse than the minimax results for standard reward-based linear MDPs (He et al., 2023).

Our algorithm and theoretical results establish a principled framework for efficient preference-based reinforcement learning with trajectory-level feedback.

### 1.1. Related Works

Among previous works, the most closely related to ours are Chen et al. (2022); Wang et al. (2023); Chen et al. (2025a).

Chen et al. (2022) learns a transition function class and plans with it to derive policies, which is extremely time- and computation-consuming. Meanwhile, Wang et al. (2023) propose a meta-algorithm that reduces the preference-based RL (PbRL) problem to a standard RL problem. When using trajectory-wise feedback, the algorithms of both approaches are *model-based*, and their results depend on the generalization of the transition kernel—specifically, its eluder dimension—which can be very large in practice.

More recently, Chen et al. (2025a) studied outcome-based reinforcement learning (i.e., using the return—sum of rewards as the learning signal) with general function approximation, which our result applies naturally by removing the comparison in our algorithm. They also extended their analysis to PbRL under the Bradley-Terry (BT) model. Their algorithm, similar to Wang et al. (2023), estimates the reward function, but relies on the return signal, and achieves a sub-optimality bound in terms of coverability (Xie et al., 2022). How-

ever, their method requires executing a composite policy that follows the optimistic policy for the first $h$ steps and a reference policy for the remaining horizon for each individual horizon $h$, which is cumbersome. This ensures that the learned reward function is step-wise accurate, whereas our algorithm operates on the entire return in a trajectory-wise manner. Moreover, their PbRL results are restricted to the BT model, whereas ours apply to more generic preference models. In addition, they do not establish bounds for linear MDPs, leaving their guarantees somewhat vague.

## 2. Preliminaries

**MDPs** We consider finite-horizon, time-inhomogeneous Markov decision processes (MDPs). Each MDP is defined by a 6-tuple $(\{\mathcal{S}_h, \mathcal{A}_h, P_h, \mathfrak{R}_h\}_{h=1}^H, s_1, H)$, where $\mathcal{S}_h$ denotes the state space, $\mathcal{A}_h$ the action space, $P_h \in (\mathcal{S}_h \times \mathcal{A}_h \to \Delta(\mathcal{S}_{h+1}))$ the transition kernel, $\mathfrak{R}_h \in (\mathcal{S}_h \times \mathcal{A}_h \to \Delta([0,1]))$ the reward distribution, $s_1$ the initial state, and $H$ the horizon.

Given a policy $\pi = \{\pi_h\}_{h=1}^H$, where $\pi_h \in (\mathcal{S}_h \to \Delta(\mathcal{A}_h))$, starting from the initial state $s_1$, we can generate a trajectory $\mathcal{T} = \{(s_h, a_h, r_h)\}_{h=1}^H$ such that $a_h \sim \pi_h(s_h)$, $r_h \sim \mathfrak{R}_h(s_h, a_h)$, and $s_{h+1} \sim P_h(\cdot|s_h, a_h)$. We denote $r_h^\star(s_h, a_h)$ as the expected reward of $(s_h, a_h)$, i.e., $r_h^\star(s_h, a_h) = \mathbb{E}_{r_h \sim \mathfrak{R}_h(s_h, a_h)} r_h$, and we will sometimes simply write it as $r_h(s_h, a_h)$ for simplicity. The value function $Q^\pi$ of a policy $\pi$ is defined as the expected cumulative rewards starting from a given state-action pair, i.e., $Q_h^\pi(s_h, a_h) = \mathbb{E}_{\{s_i, a_i\}_{i=h}^H}[\sum_{i=h}^H r_i^\star(s_i, a_i)]$.

Our objective is to find a policy $\pi$ that maximizes the expected return from the initial state $s_1$, i.e.,

$$J(\pi) := Q_1^\pi(s_1, \pi_1),$$

where

$$Q_h^\pi(s_h, \pi_h) = \mathbb{E}_{a_h \sim \pi_h(s_h)}[Q_h^\pi(s_h, a_h)].$$

We define $J^\star = J(\pi^\star), Q^\star = Q^{\pi^\star}$, where $\pi^\star$ is the optimal policy.

Given a function $Q \in \prod_{h=1}^H (\mathcal{S}_h \times \mathcal{A}_h \to [0, H])$, a policy $\pi$, and a transition kernel $P \in \prod_{h=1}^H (\mathcal{S}_h \times \mathcal{A}_h \to \Delta(\mathcal{S}_{h+1}))$, we define the backup operator $P^\pi$ as

$$(P^\pi Q)_h(s_h, a_h) := \\ \mathbb{E}_{s_{h+1} \sim P_h(s_h, a_h), a_{h+1} \sim \pi_{h+1}(s_{h+1})} Q_{h+1}(s_{h+1}, a_{h+1}).$$

We also define the greedy operator $P^\star$ as $P^\star Q = P^{\pi_Q} Q$, where $\pi_Q$ is the greedy policy w.r.t. $Q$. In what follows, we may omit the subscript $h$ when it is clear from the context.

**PbRL** In the preference-based reinforcement learning (PbRL) setting, we are given $T$, the total number of interaction rounds the agent performs. At each round $t \in [T]$, the agent proposes a policy $\pi^t$, which is executed to produce a *reward-free* trajectory $\mathcal{T}^t = \{(s_h^t, a_h^t)\}_{h=1}^H$, where $a_h^t \sim \pi^t(s_h^t)$ and $s_{h+1}^t \sim P_h(\cdot|s_h^t, a_h^t)$ for all $h \in [H]$.

After execution, the agent selects a trajectory $\mathcal{T}'$ from the set of all executed trajectories to query a preference $\widehat{p}(\mathcal{T}, \mathcal{T}')$, where $\widehat{p}(\mathcal{T}, \mathcal{T}')$ is sampled from a Bernoulli distribution with mean equal to the true preference, i.e., $\widehat{p}(\mathcal{T}, \mathcal{T}') \sim \mathrm{Bern}(p(\mathcal{T}, \mathcal{T}'))$. The goal is to minimize the cumulative regret over $T$ rounds, defined as

$$\mathrm{Regret}_T = \sum_{t=1}^T \left( J^\star - J(\pi^t) \right).$$

*Remark* 2.1. Some PbRL formulations require executing two policies at each step, which is a stricter setting compared to ours.

*Remark* 2.2. The PbRL framework naturally generalizes two well-established settings in the literature: (i) dueling bandits, which represent a direct special case of PbRL with a horizon of $H = 1$; and (ii) standard outcome-based RL, where explicit scalar returns can be used to induce preference signals.

We assume the existence of a link function $p \in ([0, H] \times [0, H]) \to [0, 1]$, and, for convenience, we slightly abuse notation by also denoting it as $p$, such that

$$p(\mathcal{T}^1, \mathcal{T}^2) = p(R^\star(\mathcal{T}^1), R^\star(\mathcal{T}^2)),$$

where $R^\star(\mathcal{T})$ denotes the expected return of trajectory $\mathcal{T}$. The link function $p$ is assumed to be known, as is standard in most PbRL literature.

**Assumption 2.3.** The link function satisfies

$$\begin{aligned}
\kappa|(R_1 - R_2) - (R_3 - R_4)| \leq &|p(R_1, R_2) - p(R_3, R_4)| \\
&\leq \kappa'|(R_1 - R_2) - (R_3 - R_4)|,
\end{aligned}$$

where $\kappa, \kappa' > 0$ are absolute constants.

*Remark* 2.4. This assumption generalizes many commonly used conditions on link functions in the preference learning literature. In particular, any link function of the form $p(R_1, R_2) = \phi(R_1 - R_2)$, where $\phi : \mathbb{R} \to [0, 1]$ is continuously differentiable with derivative bounded away from zero and infinity, satisfies Assumption 2.3. A canonical example is the Bradley–Terry (BT) model.

**Function Approximation** In this paper, we consider the setting of function approximation to achieve greater generality. In such settings, it is standard to assume the existence of function classes that satisfy certain structural properties. Specifically, we make the following assumptions.

**Assumption 2.5** (Realizability). We are given a value function class $\mathcal{Q} \subseteq \prod_{h=1}^H(\mathcal{S}_h \times \mathcal{A}_h \to [0, H])$ and a return function class $\mathcal{R} \subseteq (\prod_{h=1}^H(\mathcal{S}_h \times \mathcal{A}_h) \to [0, H])$ such that there exist $Q^\sharp \in \mathcal{Q}$ and $R^\sharp \in \mathcal{R}$ satisfying $\|Q^\sharp - Q^\star\|_\infty \leq \varepsilon_{\mathrm{app}}$ and $\|R^\sharp - R^\star\|_\infty \leq \varepsilon_{\mathrm{app}}$.

*Remark* 2.6. Compared with prior works, we assume realizability of the *return function* rather than the reward function, which is technically a weaker requirement.

**Assumption 2.7** (Completeness). We are given a backup function class $\widetilde{\mathcal{Q}} \subseteq \prod_{h=1}^H(\mathcal{S}_h \times \mathcal{A}_h \to [0, H])$ such that, for all $Q \in \mathcal{Q}$, there exists $\widetilde{Q}^\sharp(Q) \in \widetilde{\mathcal{Q}}$ satisfying $\|\widetilde{Q}^\sharp(Q) - P^\star Q\|_\infty \leq \varepsilon_{\mathrm{app}}$.

*Remark* 2.8. This notion of completeness differs slightly from the standard Bellman completeness, as it removes dependence on the reward function in the backup operator. Any function class that satisfies vanilla Bellman completeness can be transformed into such a function class by subtracting the reward function.

*Remark* 2.9. Compared with Chen et al. (2025b), instead of requiring completeness with respect to all possible rewards within a reward function class, we only require completeness with respect to the standard backup operation.

We measure the complexity of a function class using its *covering number*.

**Definition 2.10** (Covering Number). Let $\mathcal{F} \subseteq (\mathcal{X} \to \mathbb{R})$ be a class of functions and let $\alpha \geq 0$. An $\alpha$-*covering* of $\mathcal{F}$ (with respect to the $\|\cdot\|_\infty$ norm) is a subset $\mathcal{F}' \subseteq \mathcal{F}$ such that, for every $f \in \mathcal{F}$, there exists $f' \in \mathcal{F}'$ satisfying $\|f - f'\|_\infty \leq \alpha$. The $\alpha$-*covering number* of $\mathcal{F}$ is the smallest cardinality of an $\alpha$-covering of $\mathcal{F}$, defined as $\mathcal{N}(\mathcal{F}, \alpha) := \min\{ |\mathcal{F}'| : \mathcal{F}' \text{ is an } \alpha\text{-covering of } \mathcal{F} \}$.

The generalization of the function class is measured by the *Dual Episodic Eluder Dimension (DEED)*, which depends on the structure of the underlying problem. DEED builds upon the notion of the generalized eluder dimension (Agarwal et al., 2022), further incorporating both trajectory-level and comparison-based aspects of the preference, which reflect the two defining aspects of PbRL. It is defined as follows:

**Definition 2.11** (DEED for PbRL). Given a value function class $\mathcal{Q} \subseteq \prod_{h=1}^H(\mathcal{S}_h \times \mathcal{A}_h \to [0, H])$, two parameters $\lambda > 0$ and $T \in \mathbb{Z}^+$, its dual episodic eluder dimension (DEED) in PbRL is defined as

$$\mathrm{deed}_T(\mathcal{Q}; \lambda) := \max_{\{\mathcal{T}^t\}_{t=1}^T}$$

$$\sum_{t=1}^T \left[ \max_{Q \in \mathcal{Q}, Q' \in r + P^\star \mathcal{Q}} \frac{D(Q, Q'; \mathcal{T}^t, \mathcal{T}^{t-1})}{\sum_{i=1}^{t-1} D(Q, Q'; \mathcal{T}^i, \mathcal{T}^{i-1}) + \lambda} \right],$$

where $r^\star + P^\star \mathcal{Q} = \{r^\star + P^\star Q | Q \in \mathcal{Q}\}$, and

$$D(Q, Q'; \{s_h, a_h\}_{h=1}^H, \{s'_h, a'_h\}_{h=1}^H) =$$
$$\Big(\sum_{h=1}^H \big[Q_h(s_h, a_h) - Q_h(s'_h, a'_h) - Q'_h(s_h, a_h) + Q'_h(s'_h, a'_h)\big]\Big)$$

*Remark* 2.12. In the above definition, $D\big(Q, Q'; \{s_h, a_h\}_{h=1}^H, \{s'_h, a'_h\}_{h=1}^H\big)$ characterizes the degree to which the new samples are out of distribution, with this measure being governed by the imposed function class assumptions.

*Remark* 2.13 (Intuition behind DEED). Standard eluder dimensions assume additive, step-wise rewards and struggle with the relative, trajectory-level feedback inherent to PbRL. DEED addresses this by lifting the complexity measure to the space of trajectory pairs. Much like standard eluder notions, it quantifies the discrepancy between known and unknown regions of the function space. However, by measuring this over pairs, DEED directly captures the comparative performance gain, quantifying the cumulative uncertainty remaining in the preference model after observing a sequence of comparisons.

## 3. Challenges in PbRL

This section outlines the challenges posed by preference-based reinforcement learning (PbRL) and presents our solutions, emphasizing the underlying intuition. We use $\widetilde{\mathcal{O}}(\cdot)$ to hide polylogarithmic and constant factors. For a more rigorous analysis, see Section A.

In PbRL, the agent does not receive numerical reward signals. Instead, it observes *preference signals* in the form of pairwise comparisons between trajectories.

To better understand the role of preferences, let us consider: (i) what information preferences reveal, (ii) how they differ from traditional reward signals, and (iii) what challenges this difference introduces.

As a first step, we aim to reduce the problem of learning from preferences to the more familiar setting of learning from rewards. Suppose we are at step $t$ of the learning process and aim to select a good policy $\pi^t$, and we have collected a dataset

$$\mathcal{D}^{t-1} = \{(\mathcal{T}^i, \widetilde{\mathcal{T}}^i, \widehat{p}^i)\}_{i=1}^{t-1},$$

where $\widetilde{\mathcal{T}}^i = \{\widetilde{s}_h^i, \widetilde{a}_h^i\}_{h=1}^H$ is the trajectory we select to compare at step $i$.

Since preference feedback is inherently *trajectory-level*, it does not uniquely identify the underlying per-step reward function. Indeed, many different reward functions can induce the same pairwise preferences, as long as they agree on the *trajectory returns*. Consequently, what can be reliably

inferred from preferences is not the instantaneous reward, but the *cumulative reward* of a trajectory.

Motivated by this observation, we focus on learning a return function that best explains the observed preference data. A natural approach is to fit the return function by minimizing a squared loss between the predicted preference probabilities—induced by the return differences—and the observed preference signals. Formally, given a candidate return function $R$, we define the empirical preference loss as

$$\widehat{\mathcal{L}}_t^r(R) := \sum_{i=1}^{t-1} \Big(p_R(\mathcal{T}^i, \widetilde{\mathcal{T}}^i) - \widehat{p}^i\Big)^2, \tag{1}$$

where $p_R$ denotes the preference probability induced by the return function $R$.

Using this loss, we obtain the following guarantee for the learned return function.

**Lemma 3.1** (Adapted from Lemma A.6). *If Assumptions 2.3 and 2.5 hold, let $R^t = \arg\min_{R \in \mathcal{R}} \widehat{\mathcal{L}}_t^r(R)$. With high probability, for all $t \in [T]$,* [1]

$$\sum_{i=1}^{t-1} \big[(R^t - R^\star)(\mathcal{T}^i) - (R^t - R^\star)(\widetilde{\mathcal{T}}^i)\big]^2 \leq \widetilde{\mathcal{O}}\Big(\log \mathcal{N}_{\mathcal{R}}/\kappa^2\Big), \tag{2}$$

*where $\mathcal{N}_{\mathcal{R}}$ is the covering number of $\mathcal{R}$.*

*Remark* 3.2. In contrast to Wang et al. (2023); Chen et al. (2025a), we model the return instead of the per-step reward, enabling a focus on trajectory-wise signals.

*Remark* 3.3. Note that the guarantee in Lemma 3.1 is stated in terms of *differences between pairs of trajectories*, rather than absolute returns. This is inherent to preference-based feedback, which only provides information about the relative ordering of trajectories.

A natural concern arises from the fact that preference feedback is only available at the trajectory level. In particular, the contribution of a favorable action taken at a single step may be obscured by unfavorable actions occurring elsewhere along the same trajectory. This raises the question of whether effective learning is possible at all—especially for value-based methods, which rely on accurately attributing feedback to individual state–action pairs.

In the remainder of this section, we show that such learning is indeed possible, albeit in a less direct manner. The key insight is that, by carefully resolving the statistical dependence induced by trajectory-level feedback, one can still obtain reliable value estimates. Our approach decouples the correlation structure inherent in preference observations, enabling value-based learning despite the coarse nature of the feedback.

---

[1] $\widetilde{\mathcal{O}}$ ignores logarithmic and constant factors.

In particular, this section focuses on the two key challenges in value-based algorithms from the preference signal:

1. **Correlation of value function estimation.** Because the supervision comes at the trajectory level, RL methods must handle correlations that arise when decomposing trajectory-level signals into state or action values.

2. **Comparison requirement.** The feedback signal is only available in the form of comparisons. This necessitates explicit pairwise evaluation and utilization in the proof, which can be restrictive and potentially inefficient compared to scalar reward feedback.

We discuss these challenges in detail in the following subsections.

### 3.1. Correlation inside the Trajectory

Given a reward-based dataset defined as $\widetilde{\mathcal{D}}^{t-1} = \{(s_h^i, a_h^i, r_h^i)\}_{h=\in[H], i\in[t-1]}$, one of the classical losses used for value function estimation is the Bellman error:

$$\mathrm{L}_h^t(\{(s_h^i, a_h^i, r_h^i, s_{h+1}^i)\}_{i=1}^{t-1}, Q) \tag{3}$$

$$= \sum_{i=1}^{t-1} \sum_{h=1}^{H} \Big[ \Big( Q_h(s_h^i, a_h^i) - r(s_h^i, a_h^i) - Q_{h+1}(s_{h+1}^i, \pi_Q) \Big)^2 \tag{4}$$

$$- \min_{Q'\in\mathcal{Q}} \Big( Q_h'(s_h^i, a_h^i) - r(s_h^i, a_h^i) - Q_{h+1}(s_{h+1}^i, \pi_Q) \Big)^2 \Big], \tag{5}$$

where the second term is introduced to mitigate the double-sampling problem (Antos et al., 2008; Chen & Jiang, 2019; Zanette et al., 2020).

Under Bellman completeness,[2] we have

$$\mathbb{E}_{\bar{s}_h^i \sim P(\cdot|s_h^i, a_h^i)} \mathrm{L}_h^t(\{(s_h^i, a_h^i, r_h^i, \bar{s}_{h+1}^i)\}_{i=1}^{t-1}, Q)$$
$$\approx \Big( Q_h(s_h^i, a_h^i) - (P^{\pi_Q} Q + r^\star)_h(s_h^i, a_h^i) \Big)^2.$$

However, since the Bellman error is defined at the per-step level, it cannot be directly evaluated or minimized in the absence of per-step reward signals. Interestingly, one can show that accurate planning is still possible even if the value function is only correct in an aggregated, trajectory-wise sense. Specifically, suppose that a candidate action—value function $Q$ satisfies

$$\sum_{i=1}^{t-1} \Bigg( \sum_{h=1}^{H} \Big( Q(s_h^i, a_h^i) - r(s_h^i, a_h^i) - (\mathbb{P}^\star Q)_h(s_h^i, a_h^i) \Big) \Bigg)^2 \tag{6}$$

being sufficiently small. Then, planning with $Q$ is still able to produce a policy with satisfactory performance.

This perspective—focusing on trajectory-wise Bellman consistency rather than step-wise accuracy—has received relatively little attention in the literature, largely due to the strong temporal correlations induced by summing Bellman errors along a trajectory. We discuss how to address this correlation issue in the following. [3]

More concretely, a natural way to approach Equation (6) is to replace the unknown immediate rewards with an estimated trajectory return. We then aggregate the differences across all horizons and measure the squared error at the trajectory level:

$$\sum_{i=1}^{t-1} \Bigg[ \Bigg( \sum_{h=1}^{H} \Big( \underbrace{Q(s_h^i, a_h^i)}_{\mathrm{I}} - r(s_h^i, a_h^i) - \underbrace{Q(s_{h+1}, \pi_Q)}_{\mathrm{II}} \Big) \Bigg)^2 \tag{7}$$

$$- \min_{Q\in\mathcal{Q}} \Big( \sum_{h=1}^{H} \Big( Q(s_h^i, a_h^i) - r(s_h^i, a_h^i) - Q(s_{h+1}, \pi_Q) \Big) \Big)^2 \Bigg]. \tag{8}$$

Here, the sum of per-step rewards can be approximated using the learned return function. This formulation allows us to evaluate trajectory-level Bellman consistency even without access to per-step rewards, providing a practical surrogate for training value-based methods from preference data. The main difficulty with this approach lies in the correlation $(s_{h+1}^i)$ between the transitions used to (i) estimate the Bellman-updated value function ($Q(s_{h+1}^i, \pi_Q)$, II of Equation (7)), and (ii) evaluate the original function ($Q(s_{h+1}^i, a_{h+1}^i)$, I of Equation (7)).

**Our Approach** To address this issue, we first explicitly estimate the backup using the following loss:

$$\widehat{\mathcal{L}}_t^b(\widetilde{Q}, Q) \coloneqq \sum_{i=1}^{t-1} \sum_{h=1}^{H} \Big( \widetilde{Q}_h(s_h^i, a_h^i) - Q_{h+1}(s_{h+1}^i, \pi_Q) \Big)^2. \tag{9}$$

For each $Q \in \mathcal{Q}$, we define

$$\mathcal{B}^t Q \coloneqq \operatorname*{arg\,min}_{\widetilde{Q}\in\widetilde{\mathcal{Q}}} \widehat{\mathcal{L}}_t^b(\widetilde{Q}, Q). \tag{10}$$

One can show that $\mathcal{B}^t Q$ obeys the following lemma.

**Lemma 3.4** (Adapted from Lemma A.7). *Suppose Assump-*

---

[2]I.e., $Q$ is closed under Bellman backup: for all $Q \in \mathcal{Q}$, $\mathcal{T}Q \in \mathcal{Q}$.

[3]For a formal justification of why a small trajectory-wise Bellman error suffices for effective planning, see Lemma A.10.

*tion 2.7 holds. With high probability,*

$$\sum_{i=1}^{t-1} \sum_{h=1}^{H} \left( (\mathcal{B}^t Q)_h(s_h^i, a_h^i) - (P^\star Q)_h(s_h^i, a_h^i) \right)^2 \quad (11)$$

$$\leq \widetilde{\mathcal{O}}(\log(\mathcal{N}_\mathcal{Q} \mathcal{N}_{\widetilde{\mathcal{Q}}}) H^3), \quad (12)$$

*where $\mathcal{N}_\mathcal{Q}$ and $\mathcal{N}_{\widetilde{\mathcal{Q}}}$ are the covering numbers of $\mathcal{Q}$ and $\widetilde{\mathcal{Q}}$.*
*Remark 3.5.* This method is no more complex than the previous approach since the previous loss also involves a min operation.

We can then approximate the value function with the loss

$$\sum_{i=1}^{t-1} \left( \sum_{h=1}^{H} \left( Q(s_h^i, a_h^i) - r(s_h^i, a_h^i) - (\mathcal{B}^t Q)_h(s_h^i, a_h^i) \right) \right)^2 \quad (13)$$

Using Equation (12), one can show that we are approximating

$$\sum_{i=1}^{t-1} \left( \sum_{h=1}^{H} \left( Q(s_h^i, a_h^i) - r(s_h^i, a_h^i) - (\mathbb{P}^\star Q)_h(s_h^i, s_h^i) \right) \right)^2 \quad (14)$$

with Equation (13).

Nevertheless, another challenge arises: the estimated return function is only accurate for computing differences of expected returns between trajectories (Equation (2)). Hence, we cannot directly replace $\sum_{h=1}^{H} r(s_h^i, a_h^i)$ with $R^t(\mathcal{T}^i)$. Instead, we use a comparison-based loss of the form

$$\widehat{\mathcal{L}}_t^v(Q, R) \coloneqq \sum_{i=1}^{t-1} \left( \sum_{h=1}^{H} \left( Q_h(s_h^i, a_h^i) - (\mathcal{B} Q)_h(s_h^i, a_h^i) \right) \right. \quad (15)$$

$$- \sum_{h=1}^{H} \left( Q_h(\widetilde{s}_h^i, \widetilde{a}_h^i) - (\mathcal{B} Q)_h(\widetilde{s}_h^i, \widetilde{a}_h^i) \right) \quad (16)$$

$$\left. + R(\{(s_h^i, a_h^i)\}_{h=1}^H) - R(\{(\widetilde{s}_h^i, \widetilde{a}_h^i)\}_{h=1}^H) \right)^2. \quad (17)$$

One can show that the approximated value function satisfies the following lemma.
**Lemma 3.6** (Adapted from Lemma A.8). *Suppose Assumptions 2.3, 2.5 and 2.7 hold. Let $\widehat{\mathcal{Q}}^t = \{Q \in \mathcal{Q} | \widehat{\mathcal{L}}_t^v(Q, R^t) \leq \beta\}$. If $\beta$ is selected properly, with high probability, for all*

*$t \in [T]$ and $Q \in \widehat{\mathcal{Q}}^t$,*

$$\sum_{i=1}^{t-1} \left( \sum_{h=1}^{H} \left( (Q_h - P^\star Q)(s_h^i, a_h^i) \right) + R^\star(\{(s_h^i, a_h^i)\}_{h=1}^H) \right. \quad (18)$$

$$\left. - \sum_{h=1}^{H} \left( (Q_h - P^\star Q)(\widetilde{s}_h^i, \widetilde{a}_h^i) \right) - R^\star(\{(\widetilde{s}_h^i, \widetilde{a}_h^i)\}_{h=1}^H) \right)^2 \quad (19)$$

$$\leq \widetilde{\mathcal{O}} \left( H^3 + \frac{1}{\kappa^2} \right). \quad (20)$$

This estimation avoids both the double-sampling problem and the trajectory correlation. The question of how to utilize the estimated function from this comparison-based loss will be addressed in the next subsection.

### 3.2. Comparison-based Signal
**Regret Decomposition** In classical value-based algorithms, we typically estimate a value function $Q^t$ at each step $t$ with a small Bellman error while maintaining optimism:

$$Q_1^t(s_1^t, \pi_{Q^t}) \geq Q_1^\star(s_1^t, \pi_\star). \quad (21)$$

This allows us to use the following inequality, which bounds the regret by the sum of the optimal Bellman errors of the estimated value functions at every step.

$$\mathrm{Regret}(T)$$
$$\lesssim \sum_{t=1}^{T} \sum_{h=1}^{H} \left( Q_h^t(s_h^t, a_h^t) - r(s_h^t, a_h^t) - P^{\pi_t} Q_{h+1}^t(s_h^t, a_h^t) \right).$$

The issue, however, is that we cannot guarantee a small optimal Bellman error, but rather only a comparison-based loss (Equation (20)). To address this, we propose recursive optimism. We first rewrite the regret with the following dual decomposition.
**Lemma 3.7** (Adapted from Lemma A.10). *Let $Q^1, Q^2 \in \mathcal{Q}$. For any deterministic policies $\pi^1, \pi^2$, and trajectories $\{(s_h^1, a_h^1)\}_{h=1}^H$ and $\{(s_h^2, a_h^2)\}_{h=1}^H$ generated by them, with*

*high probability,*

$$2Q_1^\star(s_1, a^\star) - Q_1^{\pi^1}(s_1, a_1^1) - Q_1^{\pi^2}(s_1, a_1^2)$$

$$\lesssim \sum_{h=1}^{H} \left[ \widehat{r}_h^1(s_h^1, a_h^1) - \widehat{r}_h^1(s_h^2, a_h^2) - \left[ r(s_h^1, a_h^1) - r(s_h^2, a_h^2) \right] \right]$$

$$+ \sum_{h=1}^{H} \left[ \widehat{r}_h^2(s_h^2, a_h^2) - \widehat{r}_h^2(s_h^1, a_h^1) - \left[ r(s_h^2, a_h^2) - r(s_h^1, a_h^1) \right] \right]$$

$$+ \sum_{h=1}^{H} \left[ Q_h^\star(s_h^2, \pi^\star) - Q_h^\star(s_h^2, \pi^2) - \left[ Q_h^1(s_h^2, \pi^1) - Q_h^1(s_h^2, \pi^2) \right] \right]$$

$$+ \sum_{h=1}^{H} \left[ Q_h^\star(s_h^1, \pi^\star) - Q_h^\star(s_h^1, \pi^1) - \left[ Q_h^2(s_h^1, \pi^2) - Q_h^2(s_h^1, \pi^1) \right] \right],$$

*where* $\widehat{P}^{\pi^i} Q_{h+1}(s_h^i, a_h^i) = Q_{h+1}(s_{h+1}^i, \pi^i)$, *and* $\widehat{r}_h^i(s, a) = Q_h^i(s, a) - \widehat{P}^{\pi^i} Q_{h+1}^i(s, a)$.

Taking $Q^2 = Q^{\pi^2}$, Lemma 3.7 reduces to the following lemma (see Section A.7 for the derivation).

**Lemma 3.8.** *Let* $Q^1 \in \mathcal{Q}$. *For any deterministic policies* $\pi^1$, $\pi^2$, *and trajectories* $\{(s_h^1, a_h^1)\}_{h=1}^H$ *and* $\{(s_h^2, a_h^2)\}_{h=1}^H$ *generated by them, with high probability,*

$$Q_1^\star(s_1, a^\star) - Q_1^{\pi^1}(s_1, a_1^1) \lesssim \tag{22}$$

$$\sum_{h=1}^{H} \left[ \widehat{r}_h^1(s_h^1, a_h^1) - \widehat{r}_h^1(s_h^2, a_h^2) - \left[ r(s_h^1, a_h^1) - r(s_h^2, a_h^2) \right] \right] + \tag{23}$$

$$\sum_{h=1}^{H} \left[ Q_h^\star(s_h^2, \pi^\star) - Q_h^\star(s_h^2, \pi^2) - \left[ Q_h^1(s_h^2, \pi^1) - Q_h^1(s_h^2, \pi^2) \right] \right]. \tag{24}$$

Thus, the sub-optimality of policy $\pi^1$ reduces to familiar terms: the sum of Bellman errors but in a comparison form (Equation (23)) and a term that can be removed using the rule of optimism (Equation (24)).

Finally, applying optimism is slightly tricky: it must be based on an entire *known* trajectory rather than just the initial state. At each step $t \in [T]$, any previously collected trajectory can serve as the comparison trajectory, and thus can be used as the trajectory for the optimism. To maximize data efficiency (while avoiding unnecessary complexity), we take the trajectory collected at the previous step as the comparison and select the policy as

$$Q^t = \arg\max_{Q \in \widehat{\mathcal{Q}}} \sum_h \left[ Q_h(\widetilde{s}_h^t, \pi_Q) - Q_h(\widetilde{s}_h^t, \widetilde{a}_h^t) \right]. \tag{25}$$

With the concepts introduced above, we complete the description of our algorithm (Algorithm 1).

*Remark* 3.9. A similar idea—using the trajectory from the previous step for comparison—can also be found in prior work, e.g., Wu & Sun (2024).

## 3.3. Previous Solutions

This subsection reviews how previous works in preference-based reinforcement learning with general function approximation address the challenges discussed above.

**Robust Reward Estimation** Wang et al. (2023); Chen et al. (2025a) estimate a reward function from preference (return) signals and then use it directly as a reward proxy for policy learning. Both approaches aim for robust reward estimation: either by finding algorithms that are stable under reward perturbations, or by executing a mixture of policies across horizons. This enables a reduction from preference-based (outcome-based) learning to standard reward-based learning. In contrast, our method avoids reward estimation entirely. Instead, we introduce trajectory-level optimism to account for loss perturbations, which allows us to operate directly on returns rather than inferred rewards.

**Model-based Planning** Chen et al. (2022) propose to learn a transition model from trajectories and perform planning using both the learned dynamics and a learned preference function. However, this approach comes with notable drawbacks: planning can be computationally expensive, and model-based methods often require accurate transition estimation, which can be highly data-inefficient. In our work, we instead adopt a value-based approach. Notably, the robust method of Wang et al. (2023) is also model-based.

**Comparison-based Signal** To handle comparison-based preference signals, Wang et al. (2023); Chen et al. (2025a) introduce the use of a reference policy (or trajectory) and evaluate improvements relative to it. However, the effectiveness of this approach heavily depends on the quality of the reference. In contrast, we iteratively compare a policy only to its immediate predecessor, ensuring consistent improvement and enhancing robustness. Meanwhile, Chen et al. (2022) reduce the comparison problem to function-class exploitation using planned trajectories, which relies strongly on the structural assumptions of the function classes.

## 4. Algorithm

This section introduces our algorithm, emphasizing its design principles and theoretical guarantees. The algorithm itself is presented in Algorithm 1. For a more intuitive and technical discussion, see Section 3, and for a comparison with related methods, refer to Section 3.3. Detailed statements of results and their proofs can be found in Sections A and B.

**Lines 7–8** After initialization, the algorithm begins by estimating the return function using the loss defined in Equation (1), and the backup function using the loss in Equation (9). Together, these components provide a trajectory-

level estimate of the Bellman error (Equation (15)). Importantly, estimating the backup function does not incur additional computational cost compared to previous methods (e.g., Chen et al. (2025a)), since their value function estimation also relies on a $\min\max$ procedure that requires function approximation for all value functions.

**Lines 9–10** Because the feedback signal is derived from preferences, at each step we compare the current trajectory against the trajectory from the previous iteration. This strategy maximizes data efficiency and enhances the utility of collected data. Using the previously obtained estimate of the value function, we then introduce optimism. Our treatment of optimism differs from prior work in two key respects:

- Instead of selecting the value function with the highest expected return, we also down-estimate the performance of the previous trajectory.

- Selection is performed at the trajectory level, aggregating values across the entire horizon rather than relying solely on the first step.

The first property encourages exploration guided by preference signals, while the second mitigates potential estimation errors across the horizon, producing a more robust value function estimate.

**Lines 11–13** Finally, we execute the greedy policy derived from the estimated value function, query the preference between the new trajectory and the previous one, and update the dataset accordingly.

Our algorithm provides the following guarantee. For a more detailed statement, please refer to Section A.7.

**Theorem 4.1** (Simplified version of Theorem A.11). *Suppose Assumptions 2.3, 2.5 and 2.7 hold. With an appropriate choice of hyperparameters, the policies generated by RTPQ satisfy the following guarantee with high probability:*

$$\sum_{t=1}^{T}\left[Q_1^\star(s_1^t,\pi^\star)-Q_1^{\pi^t}(s_1^t,\pi^t)\right]\leq$$

$$\widetilde{\mathcal{O}}\Bigg(\sqrt{\left(H^3\log(\mathcal{N}_\mathcal{Q}\mathcal{N}_{\widetilde{\mathcal{Q}}})+\frac{1}{\kappa^2}\log(\mathcal{N}_\mathcal{R})\right)\cdot T\,\mathtt{deed}(\mathcal{Q};\lambda)}$$

$$+\sqrt{TH^4}\Bigg),$$

*where $\mathcal{N}_\mathcal{Q}$, $\mathcal{N}_{\widetilde{\mathcal{Q}}}$, and $\mathcal{N}_\mathcal{R}$ denote the covering numbers of the corresponding function classes.*

This result achieves the optimal $\sqrt{T}$ regret rate while accounting for the complexity of the function classes, the structure of the preference model, and the generalization properties of the value functions, which are captured by the dual episodic eluder dimension (DEED; Definition 2.11).

---

**Algorithm 1** Recursive Trajectory-based Preference Q-Learning (RTPQ)

---

**Require:** Function classes $\mathcal{Q}$, $\mathcal{R}$, $\widetilde{\mathcal{Q}}$; episode number $T$; parameter $\beta$
1: **for** $t=1,\dots,T$ **do**
2:    **if** $t=1$ **then**
3:       Select any value function $Q\in\mathcal{Q}$
4:       Collect trajectory $\mathcal{T}^1=\{(s_h^1,a_h^1)\}_{h=1}^H$ using policy $\pi^1=\pi_Q$
5:       Initialize dataset $\mathcal{D}_1\leftarrow\emptyset$
6:    **else**
7:       Estimate the return function:
$$\widehat{R}_t=\arg\min_{R\in\mathcal{R}}\widehat{\mathcal{L}}_t^r(R,\mathcal{D}_{t-1})$$
8:       Estimate the value function class:
$$\widehat{\mathcal{Q}}_t=\{Q\in\mathcal{Q}\mid\widehat{\mathcal{L}}_t^v(Q,\widehat{R}_t)\leq\beta\}$$
9:       Take previous-step trajectory for comparison:
$$\widetilde{\mathcal{T}}^t=\{(\widetilde{s}_h^t,\widetilde{a}_h^t)\}_{h=1}^H$$
10:       Optimistically select the value function:
$$Q^t=\arg\max_{Q\in\widehat{\mathcal{Q}}_t}\sum_h\left[Q_h(\widetilde{s}_h^t,\pi_Q)-Q_h(\widetilde{s}_h^t,\widetilde{a}_h^t)\right]$$
11:       Execute policy $\pi^t=\pi_{Q^t}$ and collect $\mathcal{T}^t$
12:       Query preference between $\mathcal{T}^t$ and $\widetilde{\mathcal{T}}^t$ and get $\widehat{p}^t=\widehat{p}(\mathcal{T}^t,\widetilde{\mathcal{T}}^t)$
13:       Update dataset: $\mathcal{D}_t=\mathcal{D}_{t-1}\cup\{(\mathcal{T}^t,\widetilde{\mathcal{T}}^t,\widehat{p}^t)\}$
14:    **end if**
15: **end for**

---

While expressing the bound in terms of DEED may appear somewhat abstract and less directly comparable, we also provide the following specialization to linear MDPs. For a more detailed explanation of the setting, please refer to Section B.

**Proposition 4.2** (Linear RL; Simplified version of Theorem B.4). *In the case of linear MDPs, with an appropriate choice of hyperparameters, the policies generated by RTPQ satisfy the following guarantee with high probability:*

$$\sum_{t=1}^{T}\left[Q_1^\star(s_1^t,\pi^\star)-Q_1^{\pi^t}(s_1^t,\pi^t)\right]$$
$$\leq\widetilde{\mathcal{O}}\Big(dH\sqrt{T}\,\max\{H^{3/2},1/\kappa\}\Big),$$

*where $d$ is the feature dimension.*

Compared with the minimax-optimal $\widetilde{\mathcal{O}}(d\sqrt{H^3T})$ regret, this is only a factor of $\widetilde{\mathcal{O}}(\max\{H,1/(\kappa\sqrt{H})\})$ worse. This

indicates that our algorithm achieves near-optimal performance, up to a factor that depends mildly on the horizon $H$ and the condition number $\kappa$, and remains efficient in high-dimensional settings.

*Remark* 4.3. We conjecture that the extra factor of $H$ in the regret bound reflects an inherent limitation of trajectory-level comparisons, rather than a loose analysis. To see why, consider that even in standard outcome-based RL—a strictly simpler setting—subtle shifts in feedback can be obscured over a long horizon. Because PbRL aggregates preference signals over entire trajectories instead of observing immediate step-wise rewards, this compounding uncertainty manifests as a fundamental structural hurdle.

## 5. Limitations and Future Directions

While this paper establishes a general regret bound for PbRL under function approximation, several limitations suggest directions for future work.

First, our analysis assumes that preference queries are collected passively according to a fixed procedure. Some prior works propose *active query strategies*, where the agent decides whether or not to query preferences at each step, balancing the cost of feedback with learning efficiency. Incorporating such adaptive querying into our theoretical framework remains an open direction.

Second, although our regret bound is derived for general function approximation, the concrete instantiation and quantitative guarantees in this paper are provided only for *linear MDPs*, which also include tabular MDPs as a special case. Extending these results to richer function classes—such as neural networks or kernels—poses both technical and computational challenges, and remains largely unexplored.

Third, our analysis focuses on the information-theoretic and statistical limits of PbRL, assuming implicit access to an empirical risk minimization (ERM) oracle. While relying on an ERM oracle is standard in theoretical RL, implementing these optimization and optimistic selection steps in practice often involves non-convex and computationally intensive training. Consequently, our results provide statistical guarantees rather than statements of computational efficiency, leaving the development of computationally tractable algorithms as an important challenge.

More broadly, our work highlights the need for theoretical frameworks that can capture the full spectrum of trajectory-level, comparison-based feedback in reinforcement learning. We hope that our analysis, particularly the *Dual Episodic Eluder Dimension*, can serve as a foundation for such developments.

## Acknowledgements

This work was partially supported by NSF (CNS-2403758), ARO (W911NF-24-1-0155), ONR (N000142412663), and AFOSR (FA9550-25-1-0318).

## Impact Statement

This paper advances the theoretical understanding of preference-based reinforcement learning by providing general regret guarantees under broad function approximation. The results are theoretical in nature and are not intended for direct deployment; we do not foresee immediate negative societal impacts beyond those commonly associated with reinforcement learning research.

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

# A. Proof of Section 4

## A.1. Helper Lemmas

**Theorem A.1** (Azuma-Hoeffding Inequality). *Suppose $(X_t)_{t\in[N]}$ is a supermartingale adapting to $(\mathcal{F}_t)_{t\in[N]}$ and*[4]

$$|X_k - X_{k-1}| \leq c_k, \tag{26}$$

*almost surely for all $k \in [N]$. Then for all positive integers $N$ and all positive real numbers $\epsilon$,*

$$P(X_N - X_0 \geq \epsilon) \leq \exp\left(\frac{-\epsilon^2}{2\sum_{k=1}^{N} c_k^2}\right).$$

**Lemma A.2** (Freedman-type inequality (See, e.g., (Beygelzimer et al., 2011))). *Let $(Z_t)_{t\leq T}$ be a real-valued martingale difference sequence adapted to filtration $\mathcal{F}_t$, and let $\mathbb{E}_t[\cdot] = \mathbb{E}[\cdot \mid \mathcal{F}_t]$. If $|Z_t| \leq R$ almost surely, then for any $\eta \in (0, \frac{1}{R})$ it holds that with probability at least $1 - \delta$,*

$$\sum_{t=1}^{T} Z_t \leq \eta \sum_{t=1}^{T} \mathbb{E}_{t-1}[Z_t^2] + \frac{\log(\delta^{-1})}{\eta}.$$

**Lemma A.3** (Performance Difference Lemma; Modified Version). *For any two policies $\pi^1$ and $\pi^2$,*

$$\mathbb{E}_{\{s_h^1\}_{h\in[H]}\sim\pi^1}\left[\sum_{h=1}^{H}\left[Q^{\pi^2}(s_h^1, \pi^2) - Q^{\pi^2}(s_h^1, \pi^1)\right]\right] = Q^{\pi^2}(s_1^1, \pi^2) - Q^{\pi^1}(s_1^1, \pi^1).$$

*Proof.* We can expand the left hand side with the following equations,

$$\mathbb{E}_{\{s_h^1\}_{h\in[H]}\sim\pi^1}\left[\sum_{h=1}^{H}\left[Q^{\pi^2}(s_h^1, \pi^2) - Q^{\pi^2}(s_h^1, \pi^1)\right]\right]$$

$$=\mathbb{E}_{\{s_h^1\}_{h\in[H]}\sim\pi^1}\left[\sum_{h=1}^{H}\left[Q^{\pi^2}(s_h^1, \pi^2)\right] - \sum_{h=1}^{H}\left[Q^{\pi^2}(s_{h+1}^1, \pi^2) + r(s_h^1, \pi^1)\right]\right]$$

$$=\mathbb{E}_{\{s_h^1\}_{h\in[H]}\sim\pi^1}\left[Q^{\pi^2}(s_1^1, \pi^2) - \sum_{h=1}^{H} r(s_h^1, \pi^1)\right]$$

$$=Q^{\pi^2}(s_1^1, \pi^2) - Q^{\pi^1}(s_1^1, \pi^1),$$

where the first equality follows from the Bellman equation: for all $h \in [H]$,

$$\mathbb{E}_{\{s_h^1\}_{h\in[H]}\sim\pi^1}Q^{\pi^2}(s_h^1, \pi^1) = \mathbb{E}_{\{s_h^1\}_{h\in[H]}\sim\pi^1}\left[Q^{\pi^2}(s_{h+1}^1, \pi^2) + r(s_h^1, \pi^1)\right].$$

This completes the proof. $\square$

## A.2. High Probability Events

**Lemma A.4** (Empirical Performance Difference Lemma, Comparison Version). *Let $\widetilde{\pi}^t$ be the policy that we use to collect $\widetilde{\mathcal{T}}^t$, with probability at least $1 - \delta$,*

$$\sum_{t=1}^{T}\left[\sum_{h=1}^{H}\left[Q^{\widetilde{\pi}^t}(s_h^t, \widetilde{\pi}^t) - Q^{\widetilde{\pi}^t}(s_h^t, \pi^t)\right] - \left[Q^{\widetilde{\pi}^t}(s_1^t, \widetilde{\pi}^t) - Q^{\pi^t}(s_1^t, \pi^t)\right]\right] \leq \sqrt{32TH^4\log(2/\delta)},$$

*and*

$$\sum_{t=1}^{T}\left[\sum_{h=1}^{H}\left[Q^{\pi^\star}(s_h^t, \pi^\star) - Q^{\pi^\star}(s_h^t, \pi^t)\right] - \left[Q^{\pi^\star}(s_1^t, \pi^\star) - Q^{\pi^t}(s_1^t, \pi^t)\right]\right] \leq \sqrt{32TH^4\log(2/\delta)}.$$

*We denote this event by $E1$.*

---

[4] $[N] = \{0, 1, 2 \ldots N\}$, $[M, N] = \{M \ldots N\}$. The supermartingale requires that $\mathbb{E}[X_t - X_s|\mathcal{F}_s] \leq 0$ a.s. whenever $t \geq s$.

*Proof.* Let $\{\pi^{3,t}\}_{t=1}^T$ be a sequence of (possibly random) policies such that each $\pi^{3,t}$ is determined after querying the preference at step $t-1$. Define

$$L_t = \sum_{h=1}^H \left[ Q^{\pi^{3,t}}(s_h^t, \pi^{3,t}) - Q^{\pi^{3,t}}(s_h^t, \pi^t) \right] - \mathbb{E}_{\{s_h\}_{h=1}^H \sim \pi^t} \left[ \sum_{h=1}^H \left[ Q^{\pi^{3,t}}(s_h, \pi^{3,t}) - Q^{\pi^{3,t}}(s_h, \pi^t) \right] \right].$$

Let $\{\mathcal{F}_t\}_{t=0}^T$ be the filtration where $\mathcal{F}_0$ is the trivial $\sigma$-algebra, and for each $t \geq 1$,

$$\mathcal{F}_t := \sigma \Big( \{\pi^i, \widetilde{\pi}^i, \pi^{3,i}\}_{i \in [t]}, \{\mathcal{T}^i\}_{i \in [t]}, \{\text{preference feedback at rounds } i \in [t]\} \Big).$$

Since $\pi^{3,t}$ is $\mathcal{F}_{t-1}$-measurable and $\{s_h^t\}_{h=1}^H$ are generated independently according to $\pi^t$, we have $\mathbb{E}[L_t \mid \mathcal{F}_{t-1}] = 0$. So $\{L_t\}_{t=1}^T$ forms a martingale difference sequence adapting to $\{\mathcal{F}_t\}_{t=0}^T$, and

$$|L_t| \leq 4H^2.$$

Applying Theorem A.1 yields that

$$P\Big( \sum_{t=1}^T L_t \geq \epsilon \Big) \leq \exp\Big( \frac{-\epsilon^2}{32TH^4} \Big).$$

Setting $\delta = \exp\Big( \frac{-\epsilon^2}{32TH^4} \Big)$ gives $\epsilon = \sqrt{32TH^4 \log(1/\delta)}$. Hence, with probability at least $1-\delta$,

$$\sum_{t=1}^T L_t \leq \sqrt{32TH^4 \log(1/\delta)}.$$

Therefore, with probability at least $1-\delta$,

$$\sum_{t=1}^T \left[ \sum_{h=1}^H \left[ Q^{\pi^{3,t}}(s_h^t, \pi^{3,t}) - Q^{\pi^{3,t}}(s_h^t, \pi^t) \right] - \left[ Q^{\pi^{3,t}}(s_1^t, \pi^{3,t}) - Q^{\pi^t}(s_1^t, \pi^t) \right] \right]$$
$$\leq \sum_{t=1}^T \left[ \mathbb{E}_{s_h \sim \pi^t} \left[ \sum_{h=1}^H \left[ Q^{\pi^{3,t}}(s_h, \pi^{3,t}) - Q^{\pi^{3,t}}(s_h, \pi^t) \right] \right] - \left[ Q^{\pi^{3,t}}(s_1^t, \pi^{3,t}) - Q^{\pi^t}(s_1^t, \pi^t) \right] \right]$$
$$+ \sqrt{32TH^4 \log(1/\delta)}$$
$$= \sqrt{32TH^4 \log(1/\delta)}. \tag{Lemma A.3}$$

Both $\{\pi^\star\}_{t=1}^T$ and $\{\widetilde{\pi}^t\}_{t=1}^T$ satisfy the conditions required for $\{\pi^{3,t}\}_{t=1}^T$. Applying the above bound with confidence $\delta/2$ to each sequence and taking a union bound yields the stated result. $\qquad\square$

**Lemma A.5** (Transition). *For any (possibly random) value functions $\{\bar{Q}^t\}_{t \in [T]}$ and policies $\{\bar{\pi}^t\}_{t \in [T]}$ such that $(\bar{Q}^t, \bar{\pi}^t)$ is determined after querying the preference at step $t-1$, with probability at least $1-\delta$:*

$$\left| \sum_{t=1}^T \sum_{h=1}^H \left[ P^{\pi^t} \bar{Q}_{h+1}^t(s_h^t, a_h^t) - \widehat{P}^{\pi^t} \bar{Q}_{h+1}^t(s_h^t, a_h^t) \right] \right| \leq \sqrt{8TH^4 \log(2/\delta)},$$

*We denote this event by $E(\{\bar{Q}^t\}_{t \in [T]}, \{\bar{\pi}^t\}_{t \in [T]})$.*

*Proof.* We first define

$$L_t = \sum_{h=1}^H \left[ P_{h+1}^{\bar{\pi}^t} \bar{Q}^t(s_h^t, a_h^t) - \widehat{P}_{h+1}^{\bar{\pi}^t} \bar{Q}^t(s_h^t, a_h^t) \right].$$

Let $\{\mathcal{F}_t\}_{t=0}^T$ be the filtration where $\mathcal{F}_0$ is the trivial $\sigma$-algebra, and for each $t \geq 1$,

$$\mathcal{F}_t := \sigma\Big(\{\bar{\pi}^i, \bar{Q}^i\}_{i\in[t]}, \{\mathcal{T}^i\}_{i\in[t]}, \{\text{preference feedback at rounds } i \in [t]\}\Big).$$

Since $\bar{\pi}^t$ and $\bar{Q}^t$ are $\mathcal{F}_{t-1}$-measurable, and $(s_h^t, a_h^t)_{h=1}^H$ is sampled according to $\bar{\pi}^t$, we have

$$\mathbb{E}[L_t \mid \mathcal{F}_{t-1}] = \mathbb{E}_{\{s_h^t, a_h^t\}_{h=1}^H \sim \pi^t}\left[\sum_{h=1}^H \left[P_{h+1}^{\pi^t}\bar{Q}^t(s_h^t, a_h^t) - \widehat{P}_{h+1}^{\bar{\pi}^t}\bar{Q}^t(s_h^t, a_h^t)\right]\right]$$

$$= \sum_{h=1}^H \mathbb{E}_{\{s_h^t, a_h^t\}_{h=1}^H \sim \bar{\pi}^t}\left[P_{h+1}^{\bar{\pi}^t}\bar{Q}^t(s_h^t, a_h^t) - \bar{Q}^t(s_{h+1}^t, \bar{\pi}^t)\right]$$

$$= 0.$$

So $\{L_t\}$ is a martingale difference sequence. Moreover,

$$|L_t| \leq 2H^2.$$

Applying Theorem A.1 yields

$$P\Big(\sum_{t=1}^T L_t \geq \epsilon\Big) \leq \exp\Big(\frac{-\epsilon^2}{8TH^4}\Big).$$

Setting $\delta = \exp\Big(\frac{-\epsilon^2}{8TH^4}\Big)$ gives $\epsilon = \sqrt{8TH^4 \log(1/\delta)}$. Therefore, with probability at least $1 - \delta$,

$$\sum_{t=1}^T L_t \leq \sqrt{8TH^4 \log(1/\delta)}.$$

Hence, with probability at least $1 - \delta$,

$$\sum_{t=1}^T \sum_{h=1}^H \left[P^{\bar{\pi}^t}\bar{Q}_{h+1}^t(s_h^t, a_h^t) - \widehat{P}^{\bar{\pi}^t}\bar{Q}_{h+1}^t(s_h^t, a_h^t)\right] \leq \sqrt{8TH^4 \log(1/\delta)}.$$

Similarly, with probability at least $1 - \delta$,

$$\sum_{t=1}^T \sum_{h=1}^H \left[P^{\bar{\pi}^t}\bar{Q}_{h+1}^t(s_h^t, a_h^t) - \widehat{P}^{\bar{\pi}^t}\bar{Q}_{h+1}^t(s_h^t, a_h^t)\right] \geq \sqrt{8TH^4 \log(1/\delta)}.$$

Applying the union bound, with probability at least $1 - \delta$,

$$\left|\sum_{t=1}^T \sum_{h=1}^H \left[P^{\bar{\pi}^t}\bar{Q}_{h+1}^t(s_h^t, a_h^t) - \widehat{P}^{\bar{\pi}^t}\bar{Q}_{h+1}^t(s_h^t, a_h^t)\right]\right| \leq \sqrt{8TH^4 \log(2/\delta)}.$$

This completes the proof. □

## A.3. Boundedness of the Return Loss ($\widehat{\mathcal{L}}_t^r$)

**Lemma A.6.** *Let $\alpha > 0$ be any positive number, if Assumptions 2.3 and 2.5 hold, then with probability at least $1 - \delta$, for all $t \in [T]$*

$$\sum_{i=1}^t \left[R^{t+1}(\mathcal{T}^i) - R^{t+1}(\widetilde{\mathcal{T}}^i) - R^\star(\mathcal{T}^i) + R^\star(\widetilde{\mathcal{T}}^i)\right]^2 \leq \frac{16}{\kappa^2}\log(\mathcal{N}(\mathcal{R}, \alpha)T/\delta) + \frac{16\kappa' t(\varepsilon_{\text{app}} + \alpha)}{\kappa^2}.$$

*Proof.* Let $\mathcal{R}'$ denote the $\alpha$-covering of $\mathcal{R}$ with cardinality $\mathcal{N}(\mathcal{R}, \alpha)$, and define

$$\mathrm{LR}_t(R) := \left( p_R(\mathcal{T}^t, \widetilde{\mathcal{T}}^t) - \widehat{p}(\mathcal{T}^t, \widetilde{\mathcal{T}}^t) \right)^2.$$

By definition,

$$\widehat{\mathcal{L}}_t^r(R) = \sum_{i=1}^{t-1} \mathrm{LR}_i(R).$$

Fix any $t \in \{1, 2, \dots, T\}$. At step $t$, for all $R \in \mathcal{R}$, we take the expectation with respect to $\widehat{p}(\mathcal{T}^t, \widetilde{\mathcal{T}}^t) \sim p(\mathcal{T}^t, \widetilde{\mathcal{T}}^t)$:

$$
\begin{aligned}
&\mathbb{E}_{\widehat{p}(\mathcal{T}^t, \widetilde{\mathcal{T}}^t) \sim p(\mathcal{T}^t, \widetilde{\mathcal{T}}^t)} \left[ \mathrm{LR}_t(R) - \mathrm{LR}_t(R^\star) \right] \\
=&\mathbb{E}_{\widehat{p}(\mathcal{T}^t, \widetilde{\mathcal{T}}^t) \sim p(\mathcal{T}^t, \widetilde{\mathcal{T}}^t)} \left[ \left( p_R(\mathcal{T}^t, \widetilde{\mathcal{T}}^t) - \widehat{p}(\mathcal{T}^t, \widetilde{\mathcal{T}}^t) \right)^2 - \left( p_{R^\star}(\mathcal{T}^t, \widetilde{\mathcal{T}}^t) - \widehat{p}(\mathcal{T}^t, \widetilde{\mathcal{T}}^t) \right)^2 \right] \\
=&\mathbb{E}_{\widehat{p}(\mathcal{T}^t, \widetilde{\mathcal{T}}^t) \sim p(\mathcal{T}^t, \widetilde{\mathcal{T}}^t)} \Big[ \left( p_R(\mathcal{T}^t, \widetilde{\mathcal{T}}^t) - \widehat{p}(\mathcal{T}^t, \widetilde{\mathcal{T}}^t) + p_{R^\star}(\mathcal{T}^t, \widetilde{\mathcal{T}}^t) - \widehat{p}(\mathcal{T}^t, \widetilde{\mathcal{T}}^t) \right) \\
&\qquad \cdot \left( p_R(\mathcal{T}^t, \widetilde{\mathcal{T}}^t) - p_{R^\star}(\mathcal{T}^t, \widetilde{\mathcal{T}}^t) \right) \Big] \\
=&\left[ p_R(\mathcal{T}^t, \widetilde{\mathcal{T}}^t) - p_{R^\star}(\mathcal{T}^t, \widetilde{\mathcal{T}}^t) \right]^2.
\end{aligned}
$$

Next, we consider the variance with respect to $\widehat{p}(\mathcal{T}^t, \widetilde{\mathcal{T}}^t) \sim p(\mathcal{T}^t, \widetilde{\mathcal{T}}^t)$:

$$
\begin{aligned}
&\mathrm{Var}\left( \mathrm{LR}_t(R) - \mathrm{LR}_t(R^\star) \right) \\
=&\mathbb{E}_{\widehat{p}(\mathcal{T}^t, \widetilde{\mathcal{T}}^t) \sim p(\mathcal{T}^t, \widetilde{\mathcal{T}}^t)} \left[ \left( \mathrm{LR}_t(R) - \mathrm{LR}_t(R^\star) \right)^2 \right] - \left( \mathbb{E}_{\widehat{p}(\mathcal{T}^t, \widetilde{\mathcal{T}}^t) \sim p(\mathcal{T}^t, \widetilde{\mathcal{T}}^t)} \left[ \mathrm{LR}_t(R) - \mathrm{LR}_t(R^\star) \right] \right)^2 \\
\leq&\mathbb{E}_{\widehat{p}(\mathcal{T}^t, \widetilde{\mathcal{T}}^t) \sim p(\mathcal{T}^t, \widetilde{\mathcal{T}}^t)} \left[ \left( \mathrm{LR}_t(R) - \mathrm{LR}_t(R^\star) \right)^2 \right] \\
=&\mathbb{E}_{\widehat{p}(\mathcal{T}^t, \widetilde{\mathcal{T}}^t) \sim p(\mathcal{T}^t, \widetilde{\mathcal{T}}^t)} \Big[ \left( p_R(\mathcal{T}^t, \widetilde{\mathcal{T}}^t) - \widehat{p}(\mathcal{T}^t, \widetilde{\mathcal{T}}^t) + p_{R^\star}(\mathcal{T}^t, \widetilde{\mathcal{T}}^t) - \widehat{p}(\mathcal{T}^t, \widetilde{\mathcal{T}}^t) \right)^2 \\
&\qquad \cdot \left( p_R(\mathcal{T}^t, \widetilde{\mathcal{T}}^t) - p_{R^\star}(\mathcal{T}^t, \widetilde{\mathcal{T}}^t) \right)^2 \Big] \\
\leq&4 \left( p_R(\mathcal{T}^t, \widetilde{\mathcal{T}}^t) - p_{R^\star}(\mathcal{T}^t, \widetilde{\mathcal{T}}^t) \right)^2 \\
=&4\mathbb{E}_{\widehat{p}(\mathcal{T}^t, \widetilde{\mathcal{T}}^t) \sim p(\mathcal{T}^t, \widetilde{\mathcal{T}}^t)} \left[ \mathrm{LR}_t(R) - \mathrm{LR}_t(R^\star) \right].
\end{aligned}
$$

Let $X_{i-1}$ denote the random variable containing all information available at step $i$ before observing the preference, i.e.,

$$X_{i-1} = \left( \mathcal{D}_{i-1}, \{(s_h^i, a_h^i)\}_{h=1}^H \right).$$

Using the $\sigma$-algebras generated by $\{X_i\}_{i=1}^t$ as a filtration, Lemma A.2 implies that, with probability at least $1 - \delta$,

$$
\begin{aligned}
&\sum_{i=1}^t \left[ \mathbb{E}_{\widehat{p}(\mathcal{T}^i, \widetilde{\mathcal{T}}^i) \sim p(\mathcal{T}^i, \widetilde{\mathcal{T}}^i)} \left[ \mathrm{LR}_i(R) - \mathrm{LR}_i(R^\star) \right] - \left[ \mathrm{LR}_i(R) - \mathrm{LR}_i(R^\star) \right] \right] \\
\leq&\eta \sum_{i=1}^t \mathbb{E}_{i-1} \left[ \mathbb{E}_{\widehat{p}(\mathcal{T}^i, \widetilde{\mathcal{T}}^i) \sim p(\mathcal{T}^i, \widetilde{\mathcal{T}}^i)} \left[ \mathrm{LR}_i(R) - \mathrm{LR}_i(R^\star) \right] - \left[ \mathrm{LR}_i(R) - \mathrm{LR}_i(R^\star) \right] \right]^2 + \frac{\log(\delta^{-1})}{\eta} \\
=&\eta \sum_{i=1}^t \mathbb{E}_{\widehat{p}(\mathcal{T}^i, \widetilde{\mathcal{T}}^i) \sim p(\mathcal{T}^i, \widetilde{\mathcal{T}}^i)} \left[ \mathbb{E}_{\widehat{p}(\mathcal{T}^i, \widetilde{\mathcal{T}}^i) \sim p(\mathcal{T}^i, \widetilde{\mathcal{T}}^i)} \left[ \mathrm{LR}_i(R) - \mathrm{LR}_i(R^\star) \right] - \left[ \mathrm{LR}_i(R) - \mathrm{LR}_i(R^\star) \right] \right]^2 + \frac{\log(\delta^{-1})}{\eta} \\
=&\eta \sum_{i=1}^t \mathrm{Var}\left( \mathrm{LR}_i(R) - \mathrm{LR}_i(R^\star) \right) + \frac{\log(\delta^{-1})}{\eta} \\
\leq&4\eta \sum_{i=1}^t \mathbb{E}_{\widehat{p}(\mathcal{T}^i, \widetilde{\mathcal{T}}^i) \sim p(\mathcal{T}^i, \widetilde{\mathcal{T}}^i)} \left[ \mathrm{LR}_i(R) - \mathrm{LR}_i(R^\star) \right] + \frac{\log(\delta^{-1})}{\eta},
\end{aligned}
$$

where $\eta \in (0, 1/2)$, since

$$\left| \mathbb{E}_{\widehat{p}(\mathcal{T}^i, \widetilde{\mathcal{T}}^i) \sim p(\mathcal{T}^i, \widetilde{\mathcal{T}}^i)} \left[ \mathrm{LR}_i(R) - \mathrm{LR}_i(R^\star) \right] - \left[ \mathrm{LR}_i(R) - \mathrm{LR}_i(R^\star) \right] \right| \leq 2.$$

Setting $\eta = 1/8$ yields that, for any $R \in \mathcal{R}'$, with probability at least $1 - \delta$,

$$\frac{1}{2} \sum_{i=1}^{t} \mathbb{E}_{\widehat{p}(\mathcal{T}^i, \widetilde{\mathcal{T}}^i) \sim p(\mathcal{T}^i, \widetilde{\mathcal{T}}^i)} \left[ \mathrm{LR}_i(R) - \mathrm{LR}_i(R^\star) \right] \leq \sum_{i=1}^{t} \left[ \mathrm{LR}_i(R) - \mathrm{LR}_i(R^\star) \right] + 8 \log(\delta^{-1}).$$

Applying a union bound over $R \in \mathcal{R}'$ gives that, with probability at least $1 - \delta$, for all $R \in \mathcal{R}'$,

$$\sum_{i=1}^{t} \mathbb{E}_{\widehat{p}(\mathcal{T}^i, \widetilde{\mathcal{T}}^i) \sim p(\mathcal{T}^i, \widetilde{\mathcal{T}}^i)} \left[ \mathrm{LR}_i(R) - \mathrm{LR}_i(R^\star) \right] \leq 2 \sum_{i=1}^{t} \left[ \mathrm{LR}_i(R) - \mathrm{LR}_i(R^\star) \right] + 16 \log(\mathcal{N}(\mathcal{R}, \alpha)/\delta).$$

By Assumption 2.5, let $R'^{,\sharp} \in \mathcal{R}'$ be such that

$$\| R'^{,\sharp} - R^\sharp \|_\infty \leq \alpha,$$

which implies

$$\| R'^{,\sharp} - R^\star \|_\infty \leq \alpha + \varepsilon_{\mathrm{app}}.$$

Then

$$
\begin{aligned}
\min_{R \in \mathcal{R}'} \sum_{i=1}^{t} \left[ \mathrm{LR}_i(R) - \mathrm{LR}_i(R^\star) \right] &\leq \sum_{i=1}^{t} \left[ \mathrm{LR}_i(R'^{,\sharp}) - \mathrm{LR}_i(R^\star) \right] \\
&= \sum_{i=1}^{t} \Big[ \big( p_{R'^{,\sharp}}(\mathcal{T}^i, \widetilde{\mathcal{T}}^i) - \widehat{p}(\mathcal{T}^i, \widetilde{\mathcal{T}}^i) + p_{R^\star}(\mathcal{T}^i, \widetilde{\mathcal{T}}^i) - \widehat{p}(\mathcal{T}^i, \widetilde{\mathcal{T}}^i) \big) \\
&\qquad \cdot \big( p_{R'^{,\sharp}}(\mathcal{T}^i, \widetilde{\mathcal{T}}^i) - p_{R^\star}(\mathcal{T}^i, \widetilde{\mathcal{T}}^i) \big) \Big] \\
&\leq 4 \sum_{i=1}^{t} \big| p_{R'^{,\sharp}}(\mathcal{T}^i, \widetilde{\mathcal{T}}^i) - p_{R^\star}(\mathcal{T}^i, \widetilde{\mathcal{T}}^i) \big| \\
&\leq 4\kappa' \sum_{i=1}^{t} \big| R'^{,\sharp}(\mathcal{T}^i) - R'^{,\sharp}(\widetilde{\mathcal{T}}^i) - R^\star(\mathcal{T}^i) + R^\star(\widetilde{\mathcal{T}}^i) \big| \\
&\leq 8\kappa' t(\varepsilon_{\mathrm{app}} + \alpha).
\end{aligned}
$$

Therefore, with probability at least $1 - \delta$,

$$
\begin{aligned}
\sum_{i=1}^{t} \left[ p_{R^{t+1}}(\mathcal{T}^i, \widetilde{\mathcal{T}}^i) - p_{R^\star}(\mathcal{T}^i, \widetilde{\mathcal{T}}^i) \right]^2 &= \sum_{i=1}^{t} \mathbb{E}_{\widehat{p}(\mathcal{T}^i, \widetilde{\mathcal{T}}^i) \sim p(\mathcal{T}^i, \widetilde{\mathcal{T}}^i)} \left[ \mathrm{LR}_i(R^{t+1}) - \mathrm{LR}_i(R^\star) \right] \\
&\leq 2 \min_{R \in \mathcal{R}} \sum_{i=1}^{t} \left[ \mathrm{LR}_i(R) - \mathrm{LR}_i(R^\star) \right] + 16 \log(\mathcal{N}(\mathcal{R}, \alpha)/\delta) \\
&\leq 2 \min_{R \in \mathcal{R}'} \sum_{i=1}^{t} \left[ \mathrm{LR}_i(R) - \mathrm{LR}_i(R^\star) \right] + 16 \log(\mathcal{N}(\mathcal{R}, \alpha)/\delta) \\
&\leq 16 \log(\mathcal{N}(\mathcal{R}, \alpha)/\delta) + 16\kappa' t(\varepsilon_{\mathrm{app}} + \alpha).
\end{aligned}
$$

Recalling the definition of $p_R$, we obtain

$$\sum_{i=1}^{t} \left[ p_{R^{t+1}}(\mathcal{T}^i, \widetilde{\mathcal{T}}^i) - p_{R^\star}(\mathcal{T}^i, \widetilde{\mathcal{T}}^i) \right]^2 = \sum_{i=1}^{t} \left[ p(R^{t+1}(\mathcal{T}^i), R^{t+1}(\widetilde{\mathcal{T}}^i)) - p(R^\star(\mathcal{T}^i), R^\star(\widetilde{\mathcal{T}}^i)) \right]^2.$$

By Assumption 2.3, it follows that with probability at least $1 - \delta$,

$$\sum_{i=1}^{t} \kappa^2 \left[ R^{t+1}(\mathcal{T}^i) - R^{t+1}(\widetilde{\mathcal{T}}^i) - R^\star(\mathcal{T}^i) + R^\star(\widetilde{\mathcal{T}}^i) \right]^2$$

$$\leq \sum_{i=1}^{t} \left[ p(R^{t+1}(\mathcal{T}^i), R^{t+1}(\widetilde{\mathcal{T}}^i)) - p(R^\star(\mathcal{T}^i), R^\star(\widetilde{\mathcal{T}}^i)) \right]^2$$

$$\leq 16 \log(\mathcal{N}(\mathcal{R}, \alpha)/\delta) + 16 \kappa' t (\varepsilon_{\mathrm{app}} + \alpha).$$

Finally, applying a union bound over all $t \in [T]$ yields that, with probability at least $1 - \delta$, for all $t$,

$$\sum_{i=1}^{t} \left[ R^{t+1}(\mathcal{T}^i) - R^{t+1}(\widetilde{\mathcal{T}}^i) - R^\star(\mathcal{T}^i) + R^\star(\widetilde{\mathcal{T}}^i) \right]^2 \leq \frac{16}{\kappa^2} \log(\mathcal{N}(\mathcal{R}, \alpha) T/\delta) + \frac{16 \kappa' t (\varepsilon_{\mathrm{app}} + \alpha)}{\kappa^2}.$$

This completes the proof. □

### A.4. Boundedness of the Backup Loss

**Lemma A.7.** *Let $\alpha > 0$ be any positive number. Suppose Assumption 2.7 holds. Then, with probability at least $1 - \delta$, for all $t \in [1, T]$ and $Q \in \mathcal{Q}$,*

$$\sum_{i=1}^{t-1} \sum_{h=1}^{H} \left( (\mathcal{B}^t Q)_h(s_h^i, a_h^i) - (P^\star Q)_h(s_h^i, a_h^i) \right)^2 \leq 8H^2 t(\varepsilon_{\mathrm{app}} + \alpha) + 64H^3 \log(\mathcal{N}(\mathcal{Q}, \alpha) \mathcal{N}(\widetilde{\mathcal{Q}}, \alpha) HT/\delta).$$

*Proof.* Let $\mathcal{Q}'$ and $\widetilde{\mathcal{Q}}'$ be $\alpha$-covers of $\mathcal{Q}$ and $\widetilde{\mathcal{Q}}$ with cardinalities $\mathcal{N}(\mathcal{Q}, \alpha)$ and $\mathcal{N}(\widetilde{\mathcal{Q}}, \alpha)$, and define

$$\mathrm{LQ}_h^t(\widetilde{Q}, s_{h+1}) := \left( \widetilde{Q}_h(s_h^t, a_h^t) - Q_{h+1}(s_{h+1}, \pi_Q) \right)^2.$$

Then

$$\widehat{\mathcal{L}}_t^b(\widetilde{Q}, Q) := \sum_{i=1}^{t-1} \sum_{h=1}^{H} \left( \widetilde{Q}_h(s_h^i, a_h^i) - Q_{h+1}(s_{h+1}^i, \pi_Q) \right)^2$$

$$= \sum_{i=1}^{t-1} \sum_{h=1}^{H} \mathrm{LQ}_h^t(\widetilde{Q}, s_{h+1}^i).$$

Fix $Q \in \mathcal{Q}'$, $\widetilde{Q} \in \widetilde{\mathcal{Q}}'$, $h \in [H]$, and $t \in [T]$, taking expectation over the transition,

$$\mathbb{E}_{s_{h+1} \sim P_h(\cdot | s_h^t, a_h^t)} \left[ \mathrm{LQ}_h^t(\widetilde{Q}, s_{h+1}) - \mathrm{LQ}_h^t(P^\star Q, s_{h+1}) \right] \tag{27}$$

$$= \mathbb{E}_{s_{h+1} \sim P_h(\cdot | s_h^t, a_h^t)} \left[ \left( \widetilde{Q}_h(s_h^t, a_h^t) - Q_{h+1}(s_{h+1}, \pi_Q) \right)^2 - \left( (P^\star Q)_h(s_h^t, a_h^t) - Q_{h+1}(s_{h+1}, \pi_Q) \right)^2 \right] \tag{28}$$

$$= \mathbb{E}_{s_{h+1} \sim P_h(\cdot | s_h^t, a_h^t)} \left[ \left( \widetilde{Q}_h(s_h^t, a_h^t) - (P^\star Q)_h(s_h^t, a_h^t) \right) \right. \tag{29}$$

$$\left. \cdot \left( \widetilde{Q}_h(s_h^t, a_h^t) - Q_{h+1}(s_{h+1}, \pi_Q) + (P^\star Q)_h(s_h^t, a_h^t) - Q_{h+1}(s_{h+1}, \pi_Q) \right) \right] \tag{30}$$

$$= \left( \widetilde{Q}_h(s_h^t, a_h^t) - (P^\star Q)_h(s_h^t, a_h^t) \right)^2 \tag{31}$$

The variance with respect to $s_{h+1} \sim P_h(\cdot|s_h^t, a_h^t)$ is bounded by

$$\text{Var}\Big(\text{LQ}_h^t(\widetilde{Q}, s_{h+1}) - \text{LQ}_h^t(P^\star Q, s_{h+1})\Big) \tag{32}$$

$$=\mathbb{E}_{s_{h+1}\sim P_h(\cdot|s_h^t,a_h^t)}\Big[\Big(\text{LQ}_h^t(\widetilde{Q}, s_{h+1}) - \text{LQ}_h^t(P^\star Q, s_{h+1})\Big)^2\Big] - \Big(\mathbb{E}_{s_{h+1}\sim P_h(\cdot|s_h^t,a_h^t)}\Big[\text{LQ}_h^t(\widetilde{Q}, s_{h+1}) - \text{LQ}_h^t(P^\star Q, s_{h+1})\Big]\Big)^2 \tag{33}$$

$$\leq\mathbb{E}_{s_{h+1}\sim P_h(\cdot|s_h^t,a_h^t)}\Big[\Big(\text{LQ}_h^t(\widetilde{Q}, s_{h+1}) - \text{LQ}_h^t(P^\star Q, s_{h+1})\Big)^2\Big] \tag{34}$$

$$=\mathbb{E}_{s_{h+1}\sim P_h(\cdot|s_h^t,a_h^t)}\Big[\Big(\widetilde{Q}_h(s_h^t, a_h^t) - (P^\star Q)_h(s_h^t, a_h^t)\Big)^2 \tag{35}$$

$$\cdot\Big(\widetilde{Q}_h(s_h^t, a_h^t) - Q_{h+1}(s_{h+1}, \pi_Q) + (P^\star Q)_h(s_h^t, a_h^t) - Q_{h+1}(s_{h+1}, \pi_Q)\Big)^2\Big] \tag{36}$$

$$\leq 16H^2\Big(\widetilde{Q}_h(s_h^t, a_h^t) - (P^\star Q)_h(s_h^t, a_h^t)\Big)^2. \tag{37}$$

Applying Lemma A.2 with $\eta \in (0, 1/(2H))$, with probability at least $1 - \delta$,

$$\sum_{i=1}^t \Big[\mathbb{E}_{s_{h+1}\sim P_h(\cdot|s_h^i,a_h^i)}\Big[\text{LQ}_h^i(\widetilde{Q}, s_{h+1}) - \text{LQ}_h^i(P^\star Q, s_{h+1})\Big] - \Big[\text{LQ}_h^i(\widetilde{Q}, s_{h+1}^i) - \text{LQ}_h^i(P^\star Q, s_{h+1}^i)\Big]\Big]$$

$$\leq\eta\sum_{i=1}^t \mathbb{E}_{i-1}\Big[\mathbb{E}_{s_{h+1}\sim P_h(\cdot|s_h^i,a_h^i)}\Big[\text{LQ}_h^i(\widetilde{Q}, s_{h+1}) - \text{LQ}_h^i(P^\star Q, s_{h+1})\Big] - \text{LQ}_h^i(\widetilde{Q}, s_{h+1}^i) - \text{LQ}_h^i(P^\star Q, s_{h+1}^i)\Big]^2 + \frac{\log(\delta^{-1})}{\eta}$$

$$\leq\eta\sum_{i=1}^t \text{Var}\Big(\mathbb{E}_{s_{h+1}\sim P_h(\cdot|s_h^i,a_h^i)}\Big[\text{LQ}_h^i(\widetilde{Q}, s_{h+1}) - \text{LQ}_h^i(P^\star Q, s_{h+1})\Big] - \text{LQ}_h^i(\widetilde{Q}, s_{h+1}^i) - \text{LQ}_h^i(P^\star Q, s_{h+1}^i)\Big) + \frac{\log(\delta^{-1})}{\eta}$$

$$\leq 16H^2\eta\sum_{i=1}^t \Big(\widetilde{Q}_h(s_h^i, a_h^i) - (P^\star Q)_h(s_h^i, a_h^i)\Big)^2 + \frac{\log(\delta^{-1})}{\eta}. \qquad \text{(By Equation (37))}$$

Choosing $\eta = 1/(32H^2)$ and using equation 31, we obtain

$$\frac{1}{2}\sum_{i=1}^t \Big(\widetilde{Q}_h(s_h^i, a_h^i) - (P^\star Q)_h(s_h^i, a_h^i)\Big)^2$$

$$\leq\sum_{i=1}^t \Big[\text{LQ}_h^i(\widetilde{Q}, s_{h+1}) - \text{LQ}_h^i(P^\star Q, s_{h+1}^i)\Big] + 32H^2\log(\delta^{-1}).$$

Taking a union bound over $Q \in \mathcal{Q}'$, $\widetilde{Q} \in \widetilde{\mathcal{Q}}'$, $h \in [H]$, and $t \in [T]$, with probability at least $1 - \delta$, for all $Q, \widetilde{Q}, t$,

$$\frac{1}{2}\sum_{i=1}^t \sum_{h=1}^H \Big(\widetilde{Q}_h(s_h^i, a_h^i) - (P^\star Q)_h(s_h^i, a_h^i)\Big)^2$$

$$\leq\sum_{i=1}^t \sum_{h=1}^H \Big[\text{LQ}_h^i(\widetilde{Q}, s_{h+1}) - \text{LQ}_h^i(P^\star Q, s_{h+1}^i)\Big] + 32H^3\log(\mathcal{N}(\mathcal{Q}, \alpha)\mathcal{N}(\widetilde{\mathcal{Q}}, \alpha)HT/\delta).$$

Let $Q'^{,\sharp}(Q) \in \mathcal{Q}'$ be the function such that

$$\|Q'^{,\sharp}(Q) - Q^\sharp(Q)\|_\infty \leq \alpha,$$

by Assumption 2.7,

$$\|Q'^{,\sharp}(Q) - P^{\pi_Q}Q\|_\infty \leq \alpha + \varepsilon_{\text{app}}.$$

Then

$$
\min_{\widetilde{Q}' \in \widetilde{Q}'} \sum_{i=1}^{t} \sum_{h=1}^{H} \left[ \mathrm{LQ}_h^i(\widetilde{Q}', s_{h+1}^i) - \mathrm{LQ}_h^i(P^\star Q, s_{h+1}^i) \right]
$$

$$
\leq \sum_{i=1}^{t} \sum_{h=1}^{H} \left[ \mathrm{LQ}_h^i(\widetilde{Q}'^{,\sharp}(Q), s_{h+1}^i) - \mathrm{LQ}_h^i(P^\star Q, s_{h+1}^i) \right]
$$

$$
\leq \sum_{i=1}^{t} \sum_{h=1}^{H} \left( \widetilde{Q}'^{,\sharp}(Q)_h(s_h^i, a_h^i) - (P^\star Q)_h(s_h^i, a_h^i) \right)
$$

$$
\cdot \left( \widetilde{Q}'^{,\sharp}(Q)_h(s_h^i, a_h^i) - Q_{h+1}(s_{h+1}^i, \pi_Q) + (P^\star Q)_h(s_h^i, a_h^i) - Q_{h+1}(s_{h+1}^i, \pi_Q) \right)
$$

$$
\leq 4H \sum_{i=1}^{t} \sum_{h=1}^{H} \left| \widetilde{Q}'^{,\sharp}(Q)_h(s_h^i, a_h^i) - (P^\star Q)_h(s_h^i, a_h^i) \right|
$$

$$
\leq 4H^2 t(\varepsilon_{\mathrm{app}} + \alpha).
$$

Consequently, with probability at least $1 - \delta$, for all $t \in [1, T]$ and $Q \in \mathcal{Q}$,

$$
\sum_{i=1}^{t} \sum_{h=1}^{H} \left( (\mathcal{B}^{t+1} Q)_h(s_h^i, a_h^i) - (P^\star Q)_h(s_h^i, a_h^i) \right)^2
$$

$$
= \min_{\widetilde{Q} \in \widetilde{Q}} \sum_{i=1}^{t} \sum_{h=1}^{H} \left( \widetilde{Q}_h(s_h^i, a_h^i) - (P^\star Q)_h(s_h^i, a_h^i) \right)^2
$$

$$
\leq \min_{\widetilde{Q}' \in \widetilde{Q}'} \sum_{i=1}^{t} \sum_{h=1}^{H} \left( \widetilde{Q}_h(s_h^i, a_h^i) - (P^\star Q)_h(s_h^i, a_h^i) \right)^2
$$

$$
\leq 8H^2 t(\varepsilon_{\mathrm{app}} + \alpha) + 64H^3 \log(\mathcal{N}(\mathcal{Q}, \alpha) \mathcal{N}(\widetilde{\mathcal{Q}}, \alpha) HT/\delta).
$$

This completes the proof. □

## A.5. Boundedness of the Value Loss ($\widehat{\mathcal{L}}_t^v$)

**Lemma A.8.** *Suppose Assumptions 2.3, 2.5 and 2.7 hold. Taking*

$$
\beta = 8H^2 T(\varepsilon_{\mathrm{app}} + \alpha) + 64H^3 \log(\mathcal{N}(\mathcal{Q}, \alpha) \mathcal{N}(\widetilde{\mathcal{Q}}, \alpha) TH/\delta) + \frac{16}{\kappa^2} \log(\mathcal{N}(\mathcal{R}, \alpha) T/\delta) + \frac{16\kappa' T(\varepsilon_{\mathrm{app}} + \alpha)}{\kappa^2},
$$

*with probability at least $1 - 2\delta$, for all $t \in [T]$ and $Q \in \widehat{\mathcal{Q}}^t$,*

$$
\sum_{i=1}^{t-1} \Big( \sum_{h=1}^{H} \Big( Q_h(s_h^i, a_h^i) - P^{\pi_Q} Q(s_h^i, a_h^i) \Big) - \sum_{h=1}^{H} \Big( Q_h(\widetilde{s}_h^i, \widetilde{a}_h^i) - P^{\pi_Q} Q(\widetilde{s}_h^i, \widetilde{a}_h^i) \Big)
$$

$$
+ R^\star(\{(s_h^i, a_h^i)\}_{h=1}^H) - R^\star(\{(\widetilde{s}_h^i, \widetilde{a}_h^i)\}_{h=1}^H) \Big)^2
$$

$$
\leq 16H^2 T(\varepsilon_{\mathrm{app}} + \alpha) + 128H^3 \log(\mathcal{N}(\mathcal{Q}, \alpha) \mathcal{N}(\widetilde{\mathcal{Q}}, \alpha) TH/\delta) + \frac{32}{\kappa^2} \log(\mathcal{N}(\mathcal{R}, \alpha) T/\delta) + \frac{32\kappa' T(\varepsilon_{\mathrm{app}} + \alpha)}{\kappa^2}.
$$

*We denote the above event as $E2$.*

*Proof.* We can rewrite the left hand side as

$$
\begin{aligned}
&\sum_{i=1}^{t-1} \Bigg( \sum_{h=1}^{H} \Big( Q_h(s_h^i, a_h^i) - P^{\pi_Q} Q(s_h^i, a_h^i) \Big) - \sum_{h=1}^{H} \Big( Q_h(\widetilde{s}_h^i, \widetilde{a}_h^i) - P^{\pi_Q} Q(\widetilde{s}_h^i, \widetilde{a}_h^i) \Big) \\
&\qquad + R^{\star}(\{(s_h^i, a_h^i)\}_{h=1}^{H}) - R^{\star}(\{(\widetilde{s}_h^i, \widetilde{a}_h^i)\}_{h=1}^{H}) \Bigg)^2 \\
\leq{} &\sum_{i=1}^{t-1} \Bigg( \sum_{h=1}^{H} \Big( Q_h(s_h^i, a_h^i) - \mathcal{B}^t Q(s_h^i, a_h^i) \Big) - \sum_{h=1}^{H} \Big( Q_h(\widetilde{s}_h^i, \widetilde{a}_h^i) - \mathcal{B}^t Q(\widetilde{s}_h^i, \widetilde{a}_h^i) \Big) \\
&\qquad + R^t(\{(s_h^i, a_h^i)\}_{h=1}^{H}) - R^t(\{(\widetilde{s}_h^i, \widetilde{a}_h^i)\}_{h=1}^{H}) \Bigg)^2 \\
&+ \sum_{i=1}^{t-1} \Big( R^t(\{(s_h^i, a_h^i)\}_{h=1}^{H}) - R^t(\{(\widetilde{s}_h^i, \widetilde{a}_h^i)\}_{h=1}^{H}) - R^{\star}(\{(s_h^i, a_h^i)\}_{h=1}^{H}) + R^{\star}(\{(\widetilde{s}_h^i, \widetilde{a}_h^i)\}_{h=1}^{H}) \Big)^2 \\
&+ 2 \sum_{i=1}^{t-1} \sum_{h=1}^{H} \Big( (\mathcal{B}^t Q)_h(s_h^i, a_h^i) - (P^{\pi_Q} Q)_h(s_h^i, a_h^i) \Big)^2 .
\end{aligned}
$$

Applying Lemmas [A.6](#) and [A.7](#), the definition of $\widehat{\mathcal{Q}}^t$, and the above decomposition, we obtain that, with probability at least $1 - 2\delta$,

$$
\begin{aligned}
&\sum_{i=1}^{t-1} \Bigg( \sum_{h=1}^{H} \Big( Q_h(s_h^i, a_h^i) - P^{\pi_Q} Q(s_h^i, a_h^i) \Big) - \sum_{h=1}^{H} \Big( Q_h(\widetilde{s}_h^i, \widetilde{a}_h^i) - P^{\pi_Q} Q(\widetilde{s}_h^i, \widetilde{a}_h^i) \Big) \\
&\qquad + R^{\star}(\{(s_h^i, a_h^i)\}_{h=1}^{H}) - R^{\star}(\{(\widetilde{s}_h^i, \widetilde{a}_h^i)\}_{h=1}^{H}) \Bigg)^2 \\
\leq{} &8H^2 t(\varepsilon_{\mathrm{app}} + \alpha) + 64 H^3 \log(\mathcal{N}(\mathcal{Q}, \alpha) \mathcal{N}(\widetilde{\mathcal{Q}}, \alpha) T H / \delta) + \frac{16}{\kappa^2} \log(\mathcal{N}(\mathcal{R}, \alpha) T / \delta) + \frac{16 \kappa' t(\varepsilon_{\mathrm{app}} + \alpha)}{\kappa^2} + \beta \\
={} &16 H^2 T(\varepsilon_{\mathrm{app}} + \alpha) + 128 H^3 \log(\mathcal{N}(\mathcal{Q}, \alpha) \mathcal{N}(\widetilde{\mathcal{Q}}, \alpha) T H / \delta) + \frac{32}{\kappa^2} \log(\mathcal{N}(\mathcal{R}, \alpha) T / \delta) + \frac{32 \kappa' T(\varepsilon_{\mathrm{app}} + \alpha)}{\kappa^2} .
\end{aligned}
$$

This completes the proof. $\qquad\square$

**Lemma A.9.** *Suppose Assumptions [2.3](#), [2.5](#) and [2.7](#) hold, and let*

$$
\beta = 8 H^2 T(\varepsilon_{\mathrm{app}} + \alpha) + 64 H^3 \log(\mathcal{N}(\mathcal{Q}, \alpha) \mathcal{N}(\widetilde{\mathcal{Q}}, \alpha) T H / \delta) + \frac{16}{\kappa^2} \log(\mathcal{N}(\mathcal{R}, \alpha) T / \delta) + \frac{16 \kappa' T(\varepsilon_{\mathrm{app}} + \alpha)}{\kappa^2} .
$$

*With probability at least $1 - \delta$, for all $t \in [T]$, we have $Q^{\star} \in \widehat{\mathcal{Q}}^t$. We denote this event as E3.*

*Proof.* We can rewrite the loss of $Q^\star$ as

$$
\begin{aligned}
&\sum_{i=1}^{t-1} \Big( \sum_{h=1}^{H} \Big( Q_h^\star(s_h^i, a_h^i) - \mathcal{B}^t Q^\star(s_h^i, a_h^i) \Big) - \sum_{h=1}^{H} \Big( Q_h^\star(\widetilde{s}_h^i, \widetilde{a}_h^i) - \mathcal{B}^t Q^\star(\widetilde{s}_h^i, \widetilde{a}_h^i) \Big) \\
&\qquad + R^t(\{(s_h^i, a_h^i)\}_{h=1}^H) - R^t(\{(\widetilde{s}_h^i, \widetilde{a}_h^i)\}_{h=1}^H) \Big)^2 \\
\leq &\sum_{i=1}^{t-1} \Big( \sum_{h=1}^{H} \Big( Q_h^\star(s_h^i, a_h^i) - P^{\pi^\star} Q^\star(s_h^i, a_h^i) \Big) - \sum_{h=1}^{H} \Big( Q_h^\star(\widetilde{s}_h^i, \widetilde{a}_h^i) - P^{\pi^\star} Q^\star(\widetilde{s}_h^i, \widetilde{a}_h^i) \Big) \\
&\qquad + R^\star(\{(s_h^i, a_h^i)\}_{h=1}^H) - R^\star(\{(\widetilde{s}_h^i, \widetilde{a}_h^i)\}_{h=1}^H) \Big)^2 \\
&+ \sum_{i=1}^{t-1} \Big( R^t(\{(s_h^i, a_h^i)\}_{h=1}^H) - R^t(\{(\widetilde{s}_h^i, \widetilde{a}_h^i)\}_{h=1}^H) - R^\star(\{(s_h^i, a_h^i)\}_{h=1}^H) + R^\star(\{(\widetilde{s}_h^i, \widetilde{a}_h^i)\}_{h=1}^H) \Big)^2 \\
&+ 2 \sum_{i=1}^{t-1} \sum_{h=1}^{H} \Big( (\mathcal{B}^t Q^\star)_h(s_h^i, a_h^i) - (P^{\pi^\star} Q^\star)_h(s_h^i, a_h^i) \Big)^2 \\
= &\sum_{i=1}^{t-1} \Big( R^t(\{(s_h^i, a_h^i)\}_{h=1}^H) - R^t(\{(\widetilde{s}_h^i, \widetilde{a}_h^i)\}_{h=1}^H) - R^\star(\{(s_h^i, a_h^i)\}_{h=1}^H) + R^\star(\{(\widetilde{s}_h^i, \widetilde{a}_h^i)\}_{h=1}^H) \Big)^2 \\
&+ 2 \sum_{i=1}^{t-1} \sum_{h=1}^{H} \Big( (\mathcal{B}^t Q^\star)_h(s_h^i, a_h^i) - (P^{\pi^\star} Q^\star)_h(s_h^i, a_h^i) \Big)^2,
\end{aligned}
$$

where the last equation follows from the Bellman equation. Combining this with Lemmas A.6 and A.7, we conclude that

$$
\begin{aligned}
&\sum_{i=1}^{t-1} \Big( \sum_{h=1}^{H} \Big( Q_h^\star(s_h^i, a_h^i) - \widehat{P}^{i,\pi^\star} Q^\star(s_h^i, a_h^i) \Big) - \sum_{h=1}^{H} \Big( Q_h^\star(\widetilde{s}_h^i, \widetilde{a}_h^i) - \widehat{P}^{t,\pi^\star}(\widetilde{s}_h^i, \widetilde{a}_h^i) \Big) \\
&\qquad + R^t(\{(s_h^i, a_h^i)\}_{h=1}^H) - R^t(\{(\widetilde{s}_h^i, \widetilde{a}_h^i)\}_{h=1}^H) \\
\leq &8H^2 t(\varepsilon_{\text{app}} + \alpha) + 64H^3 \log(\mathcal{N}(\mathcal{Q}, \alpha) \mathcal{N}(\widetilde{\mathcal{Q}}, \alpha) TH/\delta) + \frac{16}{\kappa^2} \log(\mathcal{N}(\mathcal{R}, \alpha) T/\delta) + \frac{16\kappa' t(\varepsilon_{\text{app}} + \alpha)}{\kappa^2} = \beta.
\end{aligned}
$$

This completes the proof. $\qquad\square$

## A.6. Dual Regret Decomposition

**Lemma A.10** (Dual Regret Decomposition)**.** *Let* $Q^1 \in \prod_{h=1}^H \big( (\mathcal{S}_h \times \mathcal{A}_h) \to [0, H] \big)$ *and* $Q^2 \in \prod_{h=1}^H \big( (\mathcal{S}_h \times \mathcal{A}_h) \to [0, H] \big)$ *be any value functions, for any two deterministic policies* $\pi_1$, $\pi_2$, *and any combination of* $\{(s_h^1, a_h^1, s_h^2, a_h^2)\}_{h=1}^H$

*such that $a_h^1 = \pi_1(s_h^1)$ and $a_h^2 = \pi_2(s_h^2)$ for all $h \in [H]$, we have*

$$2Q_1^\star(s_1, a_1^\star) - Q_1^{\pi_1}(s_1, a_1^1) - Q_1^{\pi_2}(s_1, a_1^2)$$

$$= \sum_{h=1}^{H} \left[ \widehat{r}_h^1(s_h^1, a_h^1) - \widehat{r}_h^1(s_h^2, a_h^2) - \left[ r(s_h^1, a_h^1) - r(s_h^2, a_h^2) \right] \right]$$

$$+ \sum_{h=1}^{H} \left[ \widehat{r}_h^2(s_h^2, a_h^2) - \widehat{r}_h^2(s_h^1, a_h^1) - \left[ r(s_h^2, a_h^2) - r(s_h^1, a_h^1) \right] \right]$$

$$+ \sum_{h=1}^{H} \left[ Q_h^\star(s_h^2, \pi^\star) - Q_h^\star(s_h^2, \pi^2) - \left[ Q_h^1(s_h^2, \pi^1) - Q_h^1(s_h^2, \pi^2) \right] \right]$$

$$+ \sum_{h=1}^{H} \left[ Q_h^\star(s_h^1, \pi^\star) - Q_h^\star(s_h^1, \pi^1) - \left[ Q_h^2(s_h^1, \pi^2) - Q_h^2(s_h^1, \pi^1) \right] \right]$$

$$+ \sum_{h=1}^{H} \left[ \widehat{P}^{\pi_1} Q_{h+1}^{\pi_1}(s_h^1, a_h^1) - \widehat{P}^{\pi^\star} Q_{h+1}^\star(s_h^2, a_h^2) - \left[ Q_h^{\pi_1}(s_h^1, a_h^1) - Q_h^\star(s_h^2, a_h^2) \right] + \left[ r(s_h^1, a_h^1) - r(s_h^2, a_h^2) \right] \right]$$

$$+ \sum_{h=1}^{H} \left[ \widehat{P}^{\pi_2} Q_{h+1}^{\pi_2}(s_h^2, a_h^2) - \widehat{P}^{\pi^\star} Q_{h+1}^\star(s_h^1, a_h^1) - \left[ Q_h^{\pi_2}(s_h^2, a_h^2) - Q_h^\star(s_h^1, a_h^1) \right] + \left[ r(s_h^2, a_h^2) - r(s_h^1, a_h^1) \right] \right],$$

*where $\widehat{P}^{\pi_i} Q_{h+1}(s_h^i, a_h^i) = Q_{h+1}(s_{h+1}^i, \pi_i)$, and $\widehat{r}_h^i(s, a) = Q_h^i(s, a) - \widehat{P}^{\pi_i} Q_{h+1}^i(s, a)$.*

*Proof.* We prove the claim by a stepwise (recursive) decomposition of the left-hand side. The proof proceeds by introducing a finite set of auxiliary terms and then unfolding them across time steps.

We begin by rewriting the initial difference as a sum of four components to separate contributions from the two policies and their value functions:

$$2Q_1^\star(s_1, a_1^\star) - Q_1^{\pi_1}(s_1, a_1^1) - Q_1^{\pi_2}(s_1, a_1^2)$$

$$= \underbrace{\left[ Q_1^1(s_1, a_1^1) - Q_1^1(s_1, a_1^2) \right] - \left[ Q_1^{\pi_1}(s_1, a_1^1) - Q_1^\star(s_1, a_1^2) \right]}_{D_1^1} + \underbrace{\left[ Q_1^2(s_1, a_1^2) - Q_1^2(s_1, a_1^1) \right] - \left[ Q_2^{\pi_2}(s_1, a_1^2) - Q_1^\star(s_1, a_1^1) \right]}_{D_1^2}$$

$$- \underbrace{\left[ Q_1^1(s_1, a_1^1) - Q_1^1(s_1, a_1^2) + Q_1^2(s_1, a_1^2) - Q_1^2(s_1, a_1^1) \right]}_{G_1^1} + \underbrace{Q_1^\star(s_1, a_1^\star) - Q_1^\star(s_1, a_1^2) + Q_1^\star(s_1, a_1^\star) - Q_1^\star(s_1, a_1^1)}_{G_1^2}.$$

Next, we decompose $D_1^1$ by recursively introducing the next-step functions and rewards:

$$
\begin{aligned}
D_1^1 =& \left[Q_1^1(s_1, a_1^1) - Q_1^1(s_1, a_1^2)\right] - \left[Q_1^{\pi_1}(s_1, a_1^1) - Q_1^\star(s_1, a_1^2)\right] \\
=& \left[Q_1^1(s_1, a_1^1) - Q_1^1(s_1, a_1^2)\right] - \widehat{P}^{\pi_1}\left[Q_2^1(s_1, a_1^1) - Q_2^1(s_1, a_1^2)\right] \\
& + \widehat{P}^{\pi_1}\left[Q_2^1(s_1, a_1^1) - Q_2^1(s_1, a_1^2)\right] - \widehat{P}^{\pi_1}\left[Q_2^{\pi_1}(s_1, a_1^1) - Q_2^\star(s_1, a_1^2)\right] \\
& + \widehat{P}^{\pi_1}\left[Q_2^{\pi_1}(s_1, a_1^1) - Q_2^\star(s_1, a_1^2)\right] - \left[Q_1^{\pi_1}(s_1, a_1^1) - Q_1^\star(s_1, a_1^2)\right] \\
=& \left[Q_1^1(s_1, a_1^1) - Q_1^1(s_1, a_1^2)\right] - \widehat{P}^{\pi_1}\left[Q_2^1(s_1, a_1^1) - Q_2^1(s_1, a_1^2)\right] + \widehat{P}^{\pi^\star} Q_2^\star(s_1, a_1^2) - \widehat{P}^{\pi_1} Q_2^\star(s_1, a_1^2)) \\
& + \widehat{P}^{\pi_1}\left[Q_2^1(s_1, a_1^1) - Q_2^1(s_1, a_1^2)\right] - \widehat{P}^{\pi_1}\left[Q_2^{\pi_1}(s_1, a_1^1) - Q_2^\star(s_1, a_1^2)\right] \\
& + \widehat{P}^{\pi_1} Q_2^{\pi_1}(s_1, a_1^1) - \widehat{P}^{\pi^\star} Q_2^\star(s_1, a_1^2) - \left[Q_1^{\pi_1}(s_1, a_1^1) - Q_1^\star(s_1, a_1^2)\right] \\
=& \, \widehat{r}_1^1(s_1, a_1^1) - \widehat{r}_1^1(s_1, a_1^2) - \left[r(s_1, a_1^1) - r(s_1, a_1^2)\right] + \widehat{P}^{\pi^\star} Q_2^\star(s_1, a_1^2) - \widehat{P}^{\pi_1} Q_2^\star(s_1, a_1^2) \\
& + \widehat{P}^{\pi_1}\left[Q_2^1(s_1, a_1^1) - Q_2^1(s_1, a_1^2)\right] - \widehat{P}^{\pi_1}\left[Q_2^{\pi_1}(s_1, a_1^1) - Q_2^\star(s_1, a_1^2)\right] \\
& + \widehat{P}^{\pi_1} Q_2^{\pi_1}(s_1, a_1^1) - \widehat{P}^{\pi^\star} Q_2^\star(s_1, a_1^2) - \left[Q_1^{\pi_1}(s_1, a_1^1) - Q_1^\star(s_1, a_1^2)\right] + \left[r(s_1, a_1^1) - r(s_1, a_1^2)\right] \\
=& \underbrace{\widehat{r}_1^1(s_1, a_1^1) - \widehat{r}_1^1(s_1, a_1^2) - \left[r(s_1, a_1^1) - r(s_1, a_1^2)\right]}_{\Delta r_1^1} + \underbrace{\widehat{P}^{\pi^\star} Q_2^\star(s_1, a_1^2) - \widehat{P}^{\pi_1} Q_2^\star(s_1, a_1^2)}_{\text{PQ}_1^1} \\
& + \underbrace{Q_2^1(s_2^1, a_2^1) - Q_2^1(s_2^2, \pi_1) - Q_2^{\pi_1}(s_2^1, a_2^1) + Q_2^\star(s_2^2, \pi_1)}_{O_1^1} \\
& + \underbrace{\widehat{P}^{\pi_1} Q_2^{\pi_1}(s_1, a_1^1) - \widehat{P}^{\pi^\star} Q_2^\star(s_1, a_1^2) - \left[Q_1^{\pi_1}(s_1, a_1^1) - Q_1^\star(s_1, a_1^2)\right] + \left[r(s_1, a_1^1) - r(s_1, a_1^2)\right]}_{B_1^1}.
\end{aligned}
$$

Similarly, $D_1^2$ can be decomposed in the same manner:

$$
\begin{aligned}
& \left[Q_1^2(s_1, a_1^2) - Q_1^2(s_1, a_1^1)\right] - \left[Q_2^{\pi_2}(s_1, a_1^2) - Q_1^\star(s_1, a_1^1)\right] \\
=& \underbrace{\widehat{r}_1^2(s_1, a_1^2) - \widehat{r}_1^2(s_1, a_1^1) - \left[r(s_1, a_1^2) - r(s_1, a_1^1)\right]}_{\Delta r_1^2} + \underbrace{\widehat{P}^{\pi^\star} Q_2^\star(s_1, a_1^1) - \widehat{P}^{\pi_2} Q_2^\star(s_1, a_1^1)}_{\text{PQ}_1^1} \\
& + \underbrace{Q_2^2(s_2^2, a_2^2) - Q_2^2(s_2^1, \pi_2) - Q_2^{\pi_2}(s_2^2, a_2^2) + Q_2^\star(s_2^1, \pi_2)}_{O_1^2} \\
& + \underbrace{\widehat{P}^{\pi_2} Q_2^{\pi_2}(s_1, a_1^2) - \widehat{P}^{\pi^\star} Q_2^\star(s_1, a_1^1) - \left[Q_1^{\pi_2}(s_1, a_1^2) - Q_1^\star(s_1, a_1^1)\right] + \left[r(s_1, a_1^2) - r(s_1, a_1^1)\right]}_{B_1^2}.
\end{aligned}
$$

Next we combine the two $O$-terms, $O_1^1$ and $O_1^2$. A short algebraic rearrangement shows that their sum decomposes into next-step $D$-type terms plus gap terms that we denote $\text{OP}_1$ and a small corrective group $G_2^1$, which is the second step version of $G_1^1$ appearing before:

$$
\begin{aligned}
O_1^1 + O_1^2 =& Q_2^1(s_2^1, a_2^1) - Q_2^1(s_2^2, \pi_1) - \left[Q_2^{\pi_1}(s_2^1, a_2^1) - Q_2^\star(s_2^2, \pi_1)\right] \\
& + Q_2^2(s_2^2, a_2^2) - Q_2^2(s_2^1, \pi_2) - \left[Q_2^{\pi_2}(s_2^2, a_2^2) - Q_2^\star(s_2^1, \pi_2)\right] \\
=& Q_2^1(s_2^1, a_2^1) - Q_2^1(s_2^2, a_2^2) - \left[Q_2^{\pi_1}(s_2^1, a_2^1) - Q_2^\star(s_2^2, \pi_1)\right] \\
& + Q_2^2(s_2^2, a_2^2) - Q_2^2(s_2^1, a_2^1) - \left[Q_2^{\pi_2}(s_2^2, a_2^2) - Q_2^\star(s_2^1, \pi_2)\right] + Q_2^1(s_2^2, a_2^2) - Q_2^1(s_2^2, \pi_1) + Q_2^2(s_2^1, a_2^1) - Q_2^2(s_2^1, \pi_2) \\
=& \underbrace{Q_2^1(s_2^1, a_2^1) - Q_2^1(s_2^2, a_2^2) - \left[Q_2^{\pi_1}(s_2^1, a_2^1) - Q_2^\star(s_2^2, a_2^2)\right] + Q_2^2(s_2^2, a_2^2) - Q_2^2(s_2^1, a_2^1) - \left[Q_2^{\pi_2}(s_2^2, a_2^2) - Q_2^\star(s_2^1, a_2^1)\right]}_{D_2^1 + D_2^2} \\
& \underbrace{- \left[Q_2^\star(s_2^2, a_2^2) - Q_2^\star(s_2^2, \pi_1)\right] - \left[Q_2^\star(s_2^1, a_2^1) - Q_2^\star(s_2^1, \pi_2)\right]}_{\text{OP}_1} + \underbrace{Q_2^1(s_2^2, a_2^2) - Q_2^1(s_2^2, \pi_1) + Q_2^2(s_2^1, a_2^1) - Q_2^2(s_2^1, \pi_2)}_{G_2^1}
\end{aligned}
$$

The important point is that $D_2^1 + D_2^2$ are precisely the same type of differences but evaluated at horizon 2, so the same decomposition can be applied recursively.

But before that, let us take the $\text{OP}_1$ in the last line above, as well as $\text{PQ}_1^1$, and $\text{PQ}_1^2$ before and rearrange them

$$
\begin{aligned}
\text{PQ}_1^1 + \text{PQ}_1^2 + \text{OP}_1 = & \widehat{P}^{\pi^\star} Q_2^\star(s_1, a_1^2) - \widehat{P}^{\pi_1} Q_2^\star(s_1, a_1^2) + \widehat{P}^{\pi^\star} Q_2^\star(s_1, a_1^1) - \widehat{P}^{\pi_2} Q_2^\star(s_1, a_1^1) \\
& - \left[ Q_2^\star(s_2^2, a_2^2) - Q_2^\star(s_2^2, \pi_1) \right] - \left[ Q_2^\star(s_2^1, a_2^1) - Q_2^\star(s_2^1, \pi_2) \right] \\
= & Q_2^\star(s_2^2, \pi^\star) - Q_2^\star(s_2^2, \pi_1) + Q_2^\star(s_2^1, \pi^\star) - Q_2^\star(s_2^1, \pi_2) \\
& - \left[ Q_2^\star(s_2^2, a_2^2) - Q_2^\star(s_2^2, \pi_1) \right] - \left[ Q_2^\star(s_2^1, a_2^1) - Q_2^\star(s_2^1, \pi_2) \right] \\
= & Q_2^\star(s_2^2, \pi^\star) + Q_2^\star(s_2^1, \pi^\star) - Q_2^\star(s_2^2, a_2^2) - Q_2^\star(s_2^1, a_2^1) \\
= & \underbrace{ Q_2^\star(s_2^2, \pi^\star) + Q_2^\star(s_2^1, \pi^\star) - Q_2^\star(s_2^2, \pi^2) - Q_2^\star(s_2^1, \pi^1) }_{G_2^2}.
\end{aligned}
$$

Combining all the above equations,

$$
\begin{aligned}
& 2Q_1^\star(s_1, a_1^\star) - Q_1^{\pi_1}(s_1, a_1^1) - Q_1^{\pi_2}(s_1, a_1^2) \\
= & D_1^1 + D_1^2 - G_1^1 + G_1^2 \\
= & \underbrace{\Delta r_1^1 + \text{PQ}_1^1 + O_1^1 + B_1^1}_{D_1^1} + \underbrace{\Delta r_1^2 + \text{PQ}_1^2 + O_1^2 + B_1^2}_{D_1^2} + G_1^1 + G_1^2 \\
= & \underbrace{D_2^1 + D_2^2 + \text{OP}_1 + G_2^1}_{O_1^1 + O_1^2} + \Delta r_1^1 + \text{PQ}_1^1 + B_1^1 + \Delta r_1^2 + \text{PQ}_1^2 + B_1^2 + G_1^1 + G_1^2 \\
= & \underbrace{G_2^2}_{\text{OP}_1 + \text{PQ}_1^1 + \text{PQ}_1^2} + D_2^1 + D_2^2 + G_2^1 + \Delta r_1^1 + B_1^1 + \Delta r_1^2 + B_1^2 + G_1^1 + G_1^2 \\
= & \sum_{h=1}^{H} \left( G_h^1 + G_h^2 + \Delta r_h^1 + B_h^1 + \Delta r_h^2 + B_h^2 \right).
\end{aligned}
$$

Finally, we can re-organize parts of the terms, and this gives

$$
\begin{aligned}
& \sum_{h=1}^{H} \left( G_h^1 + G_h^2 + \Delta r_h^1 + \Delta r_h^2 \right) \\
= & \sum_{h=1}^{H} \left[ \widehat{r}_h^1(s_h^1, a_h^1) - \widehat{r}_h^1(s_h^2, a_h^2) - \left[ r(s_h^1, a_h^1) - r(s_h^2, a_h^2) \right] \right] \\
& + \sum_{h=1}^{H} \left[ \widehat{r}_h^2(s_h^2, a_h^2) - \widehat{r}_h^2(s_h^1, a_h^1) - \left[ r(s_h^2, a_h^2) - r(s_h^1, a_h^1) \right] \right] \\
& + \sum_{h=1}^{H} \left[ \left[ Q_h^\star(s_h^2, \pi^\star) + Q_h^\star(s_h^1, \pi^\star) \right] - \left[ Q_h^1(s_h^2, \pi^1) + Q_h^2(s_h^1, \pi^2) \right] \right] \\
& + \sum_{h=1}^{H} \left[ \left[ Q_h^1(s_h^2, \pi^2) + Q_h^2(s_h^1, \pi^1) \right] - \left[ Q_h^\star(s_h^2, \pi^2) + Q_h^\star(s_h^1, \pi^1) \right] \right] \\
= & \sum \left[ \widehat{r}_h^1(s_h^1, a_h^1) - \widehat{r}_h^1(s_h^2, a_h^2) - \left[ r(s_h^1, a_h^1) - r(s_h^2, a_h^2) \right] \right] \\
& + \sum \left[ \widehat{r}_h^2(s_h^1, a_h^1) - \widehat{r}_h^2(s_h^2, a_h^2) - \left[ r(s_h^1, a_h^1) - r(s_h^2, a_h^2) \right] \right] \\
& + \sum_h \left[ Q_h^\star(s_h^2, \pi^\star) - Q_h^\star(s_h^2, \pi^2) - \left[ Q_h^1(s_h^2, \pi^1) - Q_h^1(s_h^2, \pi^2) \right] \right] \\
& + \sum_h \left[ Q_h^\star(s_h^1, \pi^\star) - Q_h^\star(s_h^1, \pi^1) - \left[ Q_h^2(s_h^1, \pi^2) - Q_h^2(s_h^1, \pi^1) \right] \right].
\end{aligned}
$$

This completes the proof. $\qquad\square$

### A.7. Proof of Regret Bound

We first present a more detailed version of our main result.

**Theorem A.11.** *Suppose Assumptions 2.3, 2.5 and 2.7 hold. Taking*

$$\beta = 8H^2 T(\varepsilon_{\text{app}} + \alpha) + 64H^3 \log(\mathcal{N}(\mathcal{Q}, \alpha)\mathcal{N}(\widetilde{\mathcal{Q}}, \alpha)TH/\delta) + \frac{16}{\kappa^2} \log(\mathcal{N}(\mathcal{R}, \alpha)T/\delta) + \frac{16\kappa' T(\varepsilon_{\text{app}} + \alpha)}{\kappa^2},$$

*for any $\delta \in (0, 1/7)$, with probability at least $1 - 7\delta$, the policies generated by RTPQ satisfy*

$$\sum_{t=1}^{T} \left[ Q_1^\star(s_1^t, \pi^\star) - Q_1^{\pi^t}(s_1^t, \pi^t) \right] \le$$

$$\sqrt{\left( 16H^2 T(\varepsilon_{\text{app}} + \alpha) + 128H^3 \log(\mathcal{N}(\mathcal{Q}, \alpha)\mathcal{N}(\widetilde{\mathcal{Q}}, \alpha)TH/\delta) + \frac{32}{\kappa^2} \log(\mathcal{N}(\mathcal{R}, \alpha)T/\delta) + \frac{32\kappa' T(\varepsilon_{\text{app}} + \alpha)}{\kappa^2} + \lambda \right) \cdot T \mathrm{deed}_T(\mathcal{Q}; \lambda)}$$
$$+ 22\sqrt{2TH^4 \log(2/\delta)}.$$

Let $\widetilde{\pi}^t$ denote the policy used to generate the comparison trajectory at step $t$. While our ultimate goal is to bound the regret of $\pi^t$, for convenience we first analyze the regret of both $\pi^t$ and $\widetilde{\pi}^t$.

Let $\{Q^{2,t}\}_{t=1}^T$ be helper functions to be specified later. Using Lemma A.10, we decompose the regret as

$$\sum_{t=1}^{T} \left[ 2Q_1^\star(s_1^t, \pi^\star) - Q_1^{\pi^t}(s_1^t, \pi^t) - Q_1^{\widetilde{\pi}^t}(s_1^t, \widetilde{\pi}^t) \right]$$

$$= \sum_{t=1}^{T} \left[ \sum_{h=1}^{H} \left[ \widehat{r}_h^{1,t}(s_h^t, a_h^t) - \widehat{r}_h^{1,t}(\widetilde{s}_h^t, \widetilde{a}_h^t) - \left[ r(s_h^t, a_h^t) - r(\widetilde{s}_h^t, \widetilde{a}_h^t) \right] \right] \right] \tag{T1}$$

$$+ \sum_{h=1}^{H} \left[ \widehat{r}_h^{2,t}(\widetilde{s}_h^t, \widetilde{a}_h^t) - \widehat{r}_h^{2,t}(s_h^t, a_h^t) - \left[ r(\widetilde{s}_h^t, \widetilde{a}_h^t) - r(s_h^t, a_h^t) \right] \right] \tag{T2}$$

$$+ \sum_{h=1}^{H} \left[ Q_h^\star(\widetilde{s}_h^t, \pi^\star) - Q_h^\star(\widetilde{s}_h^t, \widetilde{\pi}^t) - \left[ Q_h^t(\widetilde{s}_h^t, \pi^t) - Q_h^t(\widetilde{s}_h^t, \widetilde{\pi}^t) \right] \right] \tag{T3}$$

$$+ \sum_{h=1}^{H} \left[ Q_h^\star(s_h^t, \pi^\star) - Q_h^\star(s_h^t, \pi^t) - \left[ Q_h^{2,t}(s_h^t, \widetilde{\pi}^t) - Q_h^{2,t}(s_h^t, \pi^t) \right] \right] \tag{T4}$$

$$+ \sum_{h=1}^{H} \left[ \widehat{P}^{\pi^t} Q_{h+1}^{\pi^t}(s_h^t, a_h^t) - \widehat{P}^{\pi^\star} Q_{h+1}^\star(\widetilde{s}_h^t, \widetilde{a}_h^t) - \left[ Q_h^{\pi^t}(s_h^t, a_h^t) - Q_h^\star(\widetilde{s}_h^t, \widetilde{a}_h^t) \right] + \left[ r(s_h^t, a_h^t) - r(\widetilde{s}_h^t, \widetilde{a}_h^t) \right] \right] \tag{T5}$$

$$+ \sum_{h=1}^{H} \left[ \widehat{P}^{\widetilde{\pi}^t} Q_{h+1}^{\widetilde{\pi}^t}(\widetilde{s}_h^t, \widetilde{a}_h^t) - \widehat{P}^{\pi^\star} Q_{h+1}^\star(s_h^t, a_h^t) - \left[ Q_h^{\widetilde{\pi}^t}(\widetilde{s}_h^t, \widetilde{a}_h^t) - Q_h^\star(s_h^t, a_h^t) \right] + \left[ r(\widetilde{s}_h^t, \widetilde{a}_h^t) - r(s_h^t, a_h^t) \right] \right], \tag{T6}$$

where $\widehat{r}_h^{1,t}(s_h, a_h) = Q_h^t(s_h, a_h) - \widehat{P}^{\pi^t} Q_{h+1}^t(s_h, a_h)$, and $\widehat{r}_h^{2,t}(s_h, a_h) = Q_h^{2,t}(s_h, a_h) - \widehat{P}^{\widetilde{\pi}^t} Q_{h+1}^{2,t}(s_h, a_h)$.

In the subsequent analysis, we condition on the events $E1$, $E2$, $E3$,

$$E(\{Q^{\pi^t}\}_{t \in [T]}, \{\pi^t\}_{t \in [T]}), \quad E(\{Q^{\pi^{t-1}}\}_{t \in [T]}, \{\pi^{t-1}\}_{t \in [T]}), \quad \text{and} \quad E(\{Q^\star\}_{t \in [T]}, \{\pi^\star\}_{t \in [T]}),$$

which hold jointly with probability at least $1 - 7\delta$ by the union bound.

We now bound the terms $T1 \sim T6$ separately. By the optimism inherent in the algorithm and Lemma A.9, we immediately obtain

$$T3 \le 0.$$

For $T4$, applying the empirical performance difference lemma (Lemma A.4), we can write

$$T4 = \sum_{t=1}^{T} \left[ \sum_{h=1}^{H} \left[ Q_h^{\star}(s_h^t, \pi^{\star}) - Q_h^{\star}(s_h^t, \pi^t) - \left[ Q_h^{2,t}(s_h^t, \widetilde{\pi}^t) - Q_h^{2,t}(s_h^t, \pi^t) \right] \right] \right]$$

$$\leq \sum_{t=1}^{T} \left[ Q_1^{\star}(s_1^t, \pi^{\star}) - Q_1^{\pi^t}(s_1^t, \pi^t) - \sum_{h=1}^{H} \left[ Q_h^{2,t}(s_h^t, \widetilde{\pi}^t) - Q_h^{2,t}(s_h^t, \pi^t) \right] \right] + \sqrt{32 T H^4 \log(2/\delta)}.$$

Since $Q^{2,t}$ is not used in the algorithm and Lemma A.10 allows flexibility in its choice, we may simply set $Q^{2,t} = Q^{\widetilde{\pi}^t}$. Applying Lemma A.4 again yields

$$T4 \leq \sum_{t=1}^{T} \left[ Q_1^{\star}(s_1^t, \pi^{\star}) - Q_1^{\pi^t}(s_1^t, \pi^t) - \sum_{h=1}^{H} \left[ Q_h^{2,t}(s_h^t, \widetilde{\pi}^t) - Q_h^{2,t}(s_h^t, \pi^t) \right] \right] + \sqrt{32 T H^4 \log(2/\delta)}$$

$$\leq \sum_{t=1}^{T} \left[ Q_1^{\star}(s_1^t, \pi^{\star}) - Q_1^{\pi^t}(s_1^t, \pi^t) - \left[ Q_1^{\widetilde{\pi}^t}(s_1^t, \widetilde{\pi}^t) - Q_1^{\pi^t}(s_1^t, \pi^t) \right] \right] + 2\sqrt{32 T H^4 \log(2/\delta)}$$

$$\leq \sum_{t=1}^{T} \left[ Q_1^{\star}(s_1^t, \pi^{\star}) - Q_1^{\widetilde{\pi}^t}(s_1^t, \widetilde{\pi}^t) \right] + 2\sqrt{32 T H^4 \log(2/\delta)}.$$

Terms $T2$, $T5$, and $T6$ correspond to empirical Bellman losses for three value functions. Using the high-probability events defined above and Lemma A.5, we have

$$T2 = \sum_{t=1}^{T} \sum_{h=1}^{H} \left[ \widehat{r}_h^{2,t}(\widetilde{s}_h^t, \widetilde{a}_h^t) - \widehat{r}_h^{2,t}(s_h^t, a_h^t) - \left[ r(\widetilde{s}_h^t, \widetilde{a}_h^t) - r(s_h^t, a_h^t) \right] \right]$$

$$= \sum_{t=1}^{T} \sum_{h=1}^{H} \left[ \left( Q^{\widetilde{\pi}^t} - r - \widehat{P}^{\widetilde{\pi}^t} Q^{\widetilde{\pi}^t} \right)(\widetilde{s}_h^t, \widetilde{a}_h^t) - \left( Q^{\widetilde{\pi}^t} - r - \widehat{P}^{\widetilde{\pi}^t} Q^{\widetilde{\pi}^t} \right)(s_h^t, a_h^t) \right]$$

$$\leq \sum_{t=1}^{T} \sum_{h=1}^{H} \left[ \left( Q^{\widetilde{\pi}^t} - r - P^{\widetilde{\pi}^t} Q^{\widetilde{\pi}^t} \right)(\widetilde{s}_h^t, \widetilde{a}_h^t) - \left( Q^{\widetilde{\pi}^t} - r - P^{\widetilde{\pi}^t} Q^{\widetilde{\pi}^t} \right)(s_h^t, a_h^t) \right] + 2\sqrt{8 T H^4 \log(2/\delta)}$$

$$= 2\sqrt{8 T H^4 \log(2/\delta)}.$$

Similarly,

$$T5 \leq 2\sqrt{8 T H^4 \log(2/\delta)}, \quad \text{and} \quad T6 \leq 2\sqrt{8 T H^4 \log(2/\delta)}.$$

*Remark* A.12. Note that $\widetilde{\pi}^t$ is not necessarily the greedy policy with respect to $Q^{2,t}$ (i.e., $Q^{\widetilde{\pi}^t}$).

Term $T1$ represents the trajectory-wise Bellman error of the estimated value function $Q^t$. By Lemma A.5, we can bound it

as

$$T1$$

$$= \sum_{t=1}^{T} \sum_{h=1}^{H} \left[ \widehat{r}_h^{1,t}(s_h^t, a_h^t) - \widehat{r}_h^{1,t}(\widetilde{s}_h^t, \widetilde{a}_h^t) - \left[ r(s_h^t, a_h^t) - r(\widetilde{s}_h^t, \widetilde{a}_h^t) \right] \right]$$

$$= \sum_{t=1}^{T} \sum_{h=1}^{H} \left[ (Q_h^t - \widehat{P}^{\pi^t} Q^t - r)(s_h^t, a_h^t) - (Q_h^t - \widehat{P}^{\pi^t} Q^t - r)(\widetilde{s}_h^t, \widetilde{a}_h^t) \right]$$

$$\leq \sum_{t=1}^{T} \sum_{h=1}^{H} \left[ (Q_h^t - P^{\pi^t} Q^t - r)(s_h^t, a_h^t) - (Q_h^t - P^{\pi^t} Q^t - r)(\widetilde{s}_h^t, \widetilde{a}_h^t) \right] + \sqrt{8TH^4 \log(2/\delta)} \qquad \text{(Lemma A.5)}$$

$$= \sum_{t=1}^{T} \left[ \sqrt{ \frac{ \left( \sum_{h=1}^{H} \left( (Q_h^t - P^{\pi^t} Q^t - r)(s_h^t, a_h^t) - (Q_h^t - P^{\pi^t} Q^t - r)(\widetilde{s}_h^t, \widetilde{a}_h^t) \right) \right)^2 }{ \sum_{i=1}^{t-1} \left( \sum_{h=1}^{H} \left( (Q_h^t - P^{\pi^t} Q^t - r)(s_h^i, a_h^i) - (Q_h^t - P^{\pi^t} Q^t - r)(\widetilde{s}_h^i, \widetilde{a}_h^i) \right) \right)^2 + \lambda } } \right. $$

$$\left. \cdot \sqrt{ \sum_{i=1}^{t-1} \left( \sum_{h=1}^{H} \left( (Q_h^t - P^{\pi^t} Q^t - r)(s_h^i, a_h^i) - (Q_h^t - P^{\pi^t} Q^t - r)(\widetilde{s}_h^i, \widetilde{a}_h^i) \right) \right)^2 + \lambda } \right] + \sqrt{8TH^4 \log(2/\delta)}$$

$$= \sum_{t=1}^{T} \left[ \sqrt{ \frac{ D(Q^t, P^{\pi^t} Q^t + r; \mathcal{T}^{1,t}, \mathcal{T}^{1,t-1}) }{ \sum_{i=1}^{t-1} D(Q^t, P^{\pi^t} Q^t + r; \mathcal{T}^{1,i}, \mathcal{T}^{1,i-1}) + \lambda } } \right. $$

$$\left. \cdot \sqrt{ \underbrace{ \sum_{i=1}^{t-1} \left( \sum_{h=1}^{H} \left( (Q_h^t - P^{\pi^t} Q^t - r)(s_h^i, a_h^i) - (Q_h^t - P^{\pi^t} Q^t - r)(\widetilde{s}_h^i, \widetilde{a}_h^i) \right) \right)^2 + \lambda }_{:= L_t} } \right] + \sqrt{8TH^4 \log(2/\delta)}$$

$$= \sum_{t=1}^{T} \left[ \sqrt{ \frac{ D(Q^t, P^{\pi^t} Q^t + r; \mathcal{T}^t, \mathcal{T}^{t-1}) }{ \sum_{i=1}^{t-1} D(Q^t, P^{\pi^t} Q^t + r; \mathcal{T}^i, \mathcal{T}^{i-1}) + \lambda } \cdot (L_t + \lambda) } \right] + \sqrt{8TH^4 \log(2/\delta)}$$

$$\leq \underbrace{ \sqrt{ \max_{t \in [1,T]} (L_t + \lambda) } \cdot \sum_{t=1}^{T} \left[ \sqrt{ \frac{ D(Q^t, P^{\pi^t} Q^t + r; \mathcal{T}^t, \mathcal{T}^{t-1}) }{ \sum_{i=1}^{t-1} D(Q^t, P^{\pi^t} Q^t + r; \mathcal{T}^i, \mathcal{T}^{i-1}) + \lambda } } \right] }_{T7} + \sqrt{8TH^4 \log(2/\delta)},$$

where $T7$ is the product of the per-step loss and the generalization cost, the latter being regulated by a refined Eluder dimension.

On one hand, by Lemma A.8,

$$\max_{t \in [1,T]} L_t \leq 16 H^2 T (\varepsilon_{\text{app}} + \alpha) + 128 H^3 \log(\mathcal{N}(\mathcal{Q}, \alpha) \mathcal{N}(\widetilde{\mathcal{Q}}, \alpha) TH/\delta) + \frac{32}{\kappa^2} \log(\mathcal{N}(\mathcal{R}, \alpha) T/\delta) + \frac{32 \kappa' T (\varepsilon_{\text{app}} + \alpha)}{\kappa^2}.$$

On the other hand, applying the Cauchy-Schwarz inequality yields

$$\sum_{t=1}^{T} \sqrt{ \frac{ D(Q^t, P^{\pi^t} Q^t + r; \mathcal{T}^t, \mathcal{T}^{t-1}) }{ \sum_{i=1}^{t-1} D(Q^t, P^{\pi^t} Q^t + r; \mathcal{T}^i, \mathcal{T}^{i-1}) + \lambda } }$$

$$\leq \sqrt{T} \cdot \sqrt{ \sum_{t=1}^{T} \frac{ D(Q^t, P^{\pi^t} Q^t + r; \mathcal{T}^t, \mathcal{T}^{t-1}) }{ \sum_{i=1}^{t-1} D(Q^t, P^{\pi^t} Q^t + r; \mathcal{T}^i, \mathcal{T}^{i-1}) + \lambda } }$$

$$\leq \sqrt{T \operatorname{deed}_T(\mathcal{Q}; \lambda)}.$$

Together, these imply

$T1$

$$\leq \sqrt{\left(16H^2T(\varepsilon_{\mathrm{app}}+\alpha)+128H^3\log(\mathcal{N}(\mathcal{Q},\alpha)\mathcal{N}(\widetilde{\mathcal{Q}},\alpha)TH/\delta)+\frac{32}{\kappa^2}\log(\mathcal{N}(\mathcal{R},\alpha)T/\delta)+\frac{32\kappa'T(\varepsilon_{\mathrm{app}}+\alpha)}{\kappa^2}+\lambda\right)\cdot T\mathrm{deed}_T(\mathcal{Q};\lambda)}$$
$$+\sqrt{8TH^4\log(2/\delta)}.$$

Putting everything together, we obtain that with probability at least $1-7\delta$,

$$\sum_{t=1}^{T}\left[2Q_1^\star(s_1^t,\pi^\star)-Q_1^{\pi^t}(s_1^t,\pi^t)-Q_1^{\widetilde{\pi}^t}(s_1^t,\widetilde{\pi}^t)\right]\leq$$

$$\sqrt{\left(16H^2T(\varepsilon_{\mathrm{app}}+\alpha)+128H^3\log(\mathcal{N}(\mathcal{Q},\alpha)\mathcal{N}(\widetilde{\mathcal{Q}},\alpha)TH/\delta)+\frac{32}{\kappa^2}\log(\mathcal{N}(\mathcal{R},\alpha)T/\delta)+\frac{32\kappa'T(\varepsilon_{\mathrm{app}}+\alpha)}{\kappa^2}+\lambda\right)\cdot T\mathrm{deed}_T(\mathcal{Q};\lambda)}$$
$$+\sqrt{8TH^4\log(2/\delta)}+6\sqrt{8TH^4\log(2/\delta)}$$
$$+\sum_{t=1}^{T}\left[Q_1^\star(s_1^t,\pi^\star)-Q_1^{\widetilde{\pi}^t}(s_1^t,\widetilde{\pi}^t)\right]+2\sqrt{32TH^4\log(2/\delta)}.$$

Simplifying the above equation, we conclude that with probability at least $1-7\delta$,

$$\sum_{t=1}^{T}\left[Q_1^\star(s_1^t,\pi^\star)-Q_1^{\pi^t}(s_1^t,\pi^t)\right]$$

$$\leq\sqrt{\left(16H^2T(\varepsilon_{\mathrm{app}}+\alpha)+128H^3\log(\mathcal{N}(\mathcal{Q},\alpha)\mathcal{N}(\widetilde{\mathcal{Q}},\alpha)TH/\delta)+\frac{32}{\kappa^2}\log(\mathcal{N}(\mathcal{R},\alpha)T/\delta)+\frac{32\kappa'T(\varepsilon_{\mathrm{app}}+\alpha)}{\kappa^2}+\lambda\right)\cdot T\mathrm{deed}_T(\mathcal{Q};\lambda)}$$
$$+22\sqrt{2TH^4\log(2/\delta)}.$$

This completes the proof.

## B. Linear MDPs and Their DEED

In this section, we derive a precise regret bound for PbRL with linear MDPs via DEED.

**Linear MDPs (Jin et al., 2019)**    A linear MDP assumes the existence of a known feature map $\phi=\prod_{h=1}^{H}\phi_h$ with $\phi_h:\mathcal{S}_h\times\mathcal{A}_h\to\mathbb{R}^d$, unknown concatenated (signed) measures $\boldsymbol{\mu}=\prod_{h=1}^{H}\boldsymbol{\mu}_h$ with $\boldsymbol{\mu}_h:\mathcal{S}_h\to\mathbb{R}^d$, and an unknown parameter vector $\boldsymbol{\theta}^\star=\{\boldsymbol{\theta}_h^\star\}_{h=1}^{H}$. For any $(s_h,a_h)\in\mathcal{S}_h\times\mathcal{A}_h$, the transition and reward functions satisfy

$$P(s_{h+1}\mid s_h,a_h)=\langle\boldsymbol{\phi}_h(s_h,a_h),\boldsymbol{\mu}_{h+1}(s_{h+1})\rangle,\qquad r_h^\star(s_h,a_h)=\langle\boldsymbol{\phi}_h(s_h,a_h),\boldsymbol{\theta}_h^\star\rangle.$$

We assume $\|\boldsymbol{\phi}_h(s_h,a_h)\|_2\leq 1$ and $\max\{\|\boldsymbol{\mu}_h(\mathcal{S}_h)\|_2,\|\boldsymbol{\theta}_h^\star\|_2\}\leq\sqrt{d}$ for all $(s_h,a_h)$ and $h\in[H]$. For a trajectory $\mathcal{T}=\{(s_h,a_h)\}_{h=1}^{H}$, define $\boldsymbol{\phi}(\mathcal{T})=\sum_{h=1}^{H}\boldsymbol{\phi}_h(s_h,a_h)$.

For any policy $\pi$ and any $(s_h,a_h)$,

$$Q_h^\pi(s_h,a_h)=r_h^\star(s_h,a_h)+\int_{\mathcal{S}_{h+1}}V_{h+1}^\pi(s_{h+1})\langle\boldsymbol{\phi}_h(s_h,a_h),\boldsymbol{\mu}_{h+1}(s_{h+1})\rangle\,ds_{h+1}$$
$$=\langle\boldsymbol{\phi}_h(s_h,a_h),\boldsymbol{\theta}_h^\star\rangle+\int_{\mathcal{S}_{h+1}}V_{h+1}^\pi(s_{h+1})\langle\boldsymbol{\phi}_h(s_h,a_h),\boldsymbol{\mu}_{h+1}(s_{h+1})\rangle\,ds_{h+1}.$$

Thus, for any policy $\pi$, there exist weights $\{\boldsymbol{w}_h^\pi\}_{h=1}^{H}$ such that

$$Q_h^\pi(s_h,a_h)=\langle\boldsymbol{\phi}_h(s_h,a_h),\boldsymbol{w}_h^\pi\rangle,$$

where we may write

$$\boldsymbol{w}_h^\pi = \boldsymbol{\theta}_h^\star + \int_{\mathcal{S}_{h+1}} V_{h+1}^\pi(s_{h+1})\boldsymbol{\mu}_{h+1}(s_{h+1})\,ds_{h+1}, \qquad \|\boldsymbol{w}_h^\pi\|_2 \le 2H\sqrt{d}.$$

In the discussion below, we omit the subscript $h$ when it is clear from context.

**Function Classes** Define the return function class

$$\mathcal{R} = \left\{ \langle \boldsymbol{\phi}_e(\cdot), \boldsymbol{\theta}_{[H]} \rangle \,\Big|\, \|\boldsymbol{\theta}_h\|_2 \le \sqrt{d},\ \forall h \in [H] \right\},$$

and the value/backup function class

$$\mathcal{Q} = \prod_{h=1}^H \mathcal{Q}_h, \qquad \mathcal{Q}_h = \left\{ \langle \boldsymbol{\phi}_h(\cdot, \cdot), \boldsymbol{w} \rangle \,\Big|\, \|\boldsymbol{w}\|_2 \le 2H\sqrt{d} \right\},$$

where

$$\boldsymbol{\phi}_e(\{(s_h, a_h)\}_{h=1}^H) = \begin{bmatrix} \boldsymbol{\phi}_1(s_1, a_1) \\ \boldsymbol{\phi}_2(s_2, a_2) \\ \vdots \\ \boldsymbol{\phi}_H(s_H, a_H) \end{bmatrix} \quad \text{and} \quad \boldsymbol{\theta}_{[H]} = \begin{bmatrix} \boldsymbol{\theta}_1 \\ \boldsymbol{\theta}_2 \\ \vdots \\ \boldsymbol{\theta}_H \end{bmatrix}$$

are $Hd \times 1$ column vectors. Recall the definition of $R^\star$,

$$R^\star(\mathcal{T}) = \sum_{(s_h, a_h) \in \mathcal{T}} \langle \boldsymbol{\phi}_h(s_h, a_h), \boldsymbol{\theta}_h^\star \rangle = \langle \boldsymbol{\phi}_e(\mathcal{T}), \boldsymbol{\theta}_{[H]}^\star \rangle,$$

so $R^\star \in \mathcal{R}$, and by the linearity of $Q_h^\star$, we have $Q^\star \in \mathcal{Q}$.

Regarding the completeness, let $Q_{\boldsymbol{w}} \in \mathcal{Q}$ and let $\pi_Q$ be the greedy policy induced by $Q_{\boldsymbol{w}}$. For any $(s_h, a_h) \in \mathcal{S}_h \times \mathcal{A}_h$, we have

$$(P^{\pi_Q} Q_{\boldsymbol{w}})_h(s_h, a_h) = \int_{\mathcal{S}_{h+1}} Q_{\boldsymbol{w}, h+1}(s_{h+1}, \pi_Q(s_{h+1}))\, P(s_{h+1} \mid s_h, a_h)\, ds_{h+1}$$

$$= \int_{\mathcal{S}_{h+1}} \langle \boldsymbol{w}_{h+1}, \boldsymbol{\phi}_{h+1}(s_{h+1}, \pi_Q(s_{h+1})) \rangle \langle \boldsymbol{\phi}_h(s_h, a_h), \boldsymbol{\mu}_{h+1}(s_{h+1}) \rangle\, ds_{h+1}.$$

By linearity of the inner product and Fubini's theorem, this can be rewritten as

$$(P^{\pi_Q} Q_{\boldsymbol{w}})_h(s_h, a_h) = \left\langle \boldsymbol{\phi}_h(s_h, a_h), \int_{\mathcal{S}_{h+1}} \langle \boldsymbol{w}_{h+1}, \boldsymbol{\phi}_{h+1}(s_{h+1}, \pi_Q(s_{h+1})) \rangle \boldsymbol{\mu}_{h+1}(s_{h+1})\, ds_{h+1} \right\rangle$$

$$= \langle \boldsymbol{w}_h', \boldsymbol{\phi}_h(s_h, a_h) \rangle,$$

where

$$\boldsymbol{w}_h' := \int_{\mathcal{S}_{h+1}} \langle \boldsymbol{w}_{h+1}, \boldsymbol{\phi}_{h+1}(s_{h+1}, \pi_Q(s_{h+1})) \rangle \boldsymbol{\mu}_{h+1}(s_{h+1})\, ds_{h+1}.$$

Moreover, by the Cauchy-Schwarz inequality and the assumptions $\|\boldsymbol{\phi}_{h+1}(\cdot, \cdot)\|_2 \le 1$ and $\|\boldsymbol{\mu}_{h+1}(\mathcal{S}_{h+1})\|_2 \le \sqrt{d}$, we have

$$\|\boldsymbol{w}_h'\|_2 \le \int_{\mathcal{S}_{h+1}} |\langle \boldsymbol{w}_{h+1}, \boldsymbol{\phi}_{h+1}(s_{h+1}, \pi_Q(s_{h+1})) \rangle| \, \|\boldsymbol{\mu}_{h+1}(s_{h+1})\|_2 \, ds_{h+1} \le \|\boldsymbol{w}_{h+1}\|_2 \le 2H\sqrt{d}.$$

Therefore, $(P^{\pi_Q} Q_{\boldsymbol{w}})_h \in \mathcal{Q}_h$, and hence $\mathcal{Q}$ is closed under the Bellman backup operator.

As a result, we can have the following proposition.

**Proposition B.1.** $\mathcal{R}$ *and* $\mathcal{Q}$ *satisfy Assumptions 2.5 and 2.7 with* $\varepsilon_{\mathrm{app}} = 0$.

We now turn to the covering number of the function classes. By Lemma B.5, we obtain the following bounds.

**Proposition B.2.** *The covering numbers of* $\mathcal{Q}$ *and* $\mathcal{R}$ *satisfy*

$$\mathcal{N}(\mathcal{Q}, \alpha) \leq \left(\frac{6H\sqrt{d}}{\alpha}\right)^{dH}, \qquad \mathcal{N}(\mathcal{R}, \alpha) \leq \left(\frac{3\sqrt{dH}}{\alpha}\right)^{dH}.$$

Next, we characterize the DEED complexity in linear MDPs. The proof is deferred to Section B.2.

**Lemma B.3.** *In linear MDPs, for any* $\lambda > 0$,

$$\mathtt{deed}_T(\mathcal{Q}; \lambda) \leq 2dH \log\left(\frac{\lambda + 64TH^3}{\lambda}\right).$$

Combining the above results, we can bound the regret of PbRL in linear MDPs.

**Theorem B.4.** *The regret of RTPQ satisfies*

$$\sum_{t=1}^{T} \left[Q_1^\star(s_1^t, \pi^\star) - Q_1^{\pi^t}(s_1^t, \pi^t)\right]$$

$$\leq \sqrt{\left(16H^2T\alpha + 128H^3 \log\left(\left(6H\sqrt{d}/\alpha\right)^{2dH}TH/\delta\right) + \frac{32}{\kappa^2} \log\left(\left(3\sqrt{dH}/\alpha\right)^{dH}T/\delta\right) + \frac{32\kappa' t\alpha}{\kappa^2} + \lambda\right) \cdot 2dHT \log\left(\frac{\lambda + 64TH^3}{\lambda}\right)}$$

$$\quad + 22\sqrt{2TH^4 \log(2/\delta)}$$

$$\leq \widetilde{\mathcal{O}}\left(dH\sqrt{T} \max\{H^{3/2}, 1/\kappa\}\right),$$

*where* $\widetilde{\mathcal{O}}(\cdot)$ *hides polylogarithmic factors and small constant numerical factors.*

### B.1. Helper Lemmas

**Lemma B.5.** *Let* $\mathcal{F} = \{f_{\boldsymbol{w}}(x) = \langle x, \boldsymbol{w}\rangle \mid \|\boldsymbol{w}\|_2 \leq m\}$ *be a class of linear functions defined on* $\mathcal{X} = \{x \in \mathbb{R}^n \mid \|x\|_2 \leq 1\}$. *Then, for any* $\varepsilon > 0$, *the covering number with respect to the sup-norm satisfies*

$$\mathcal{N}(\mathcal{F}, \varepsilon, \|\cdot\|_\infty) \leq \left(\frac{3m}{\varepsilon}\right)^n.$$

*Proof.* For any $\boldsymbol{w}_1, \boldsymbol{w}_2 \in \mathbb{R}^n$ with $\|\boldsymbol{w}_1\|_2 \leq m$ and $\|\boldsymbol{w}_2\|_2 \leq m$, we have

$$\max_{x \in \mathcal{X}} |f_{\boldsymbol{w}_1}(x) - f_{\boldsymbol{w}_2}(x)| = \max_{x \in \mathcal{X}} |\langle \boldsymbol{w}_1 - \boldsymbol{w}_2, x\rangle|$$

$$\leq \max_{x \in \mathcal{X}} \|x\|_2 \cdot \|\boldsymbol{w}_1 - \boldsymbol{w}_2\|_2$$

$$\leq \|\boldsymbol{w}_1 - \boldsymbol{w}_2\|_2.$$

Thus, any $\varepsilon$-cover of $\mathcal{F}$ under $\|\cdot\|_\infty$ is also an $\varepsilon$-cover of the Euclidean ball

$$B_2(m) = \{\boldsymbol{w} \in \mathbb{R}^n \mid \|\boldsymbol{w}\|_2 \leq m\}.$$

By a standard covering bound for Euclidean balls (see Vershynin (2011)),

$$\mathcal{N}(B_2(m), \varepsilon) \leq \left(m\left(1 + \frac{2}{\varepsilon}\right)\right)^n \leq \left(\frac{3m}{\varepsilon}\right)^n.$$

The result follows. $\qquad\qquad\square$

## B.2. Proof Lemma B.3

**Lemma B.6.** *In linear MDPs, for any $\lambda > 0$,*

$$\texttt{deed}_T(\mathcal{Q}; \lambda) \leq 2dH \log\left(\frac{\lambda + 64TH^3}{\lambda}\right).$$

*Proof.* Let $\{(s_h^t, a_h^t)\}_{h \in [H], t \in [T]}$ and $\{(\boldsymbol{w}^t, \widetilde{\boldsymbol{w}}^t)\}$ be the trajectory sequences and function parameters that achieve the maximum inside $\texttt{deed}$. I.e.,

$$
\begin{aligned}
&F_t(\mathcal{Q}, \lambda) \\
&= \frac{\left(\sum_{h=1}^{H}\left(\langle\boldsymbol{\phi}(s_h^t, a_h^t), \boldsymbol{w}_h^t\rangle - \langle\boldsymbol{\phi}(s_h^t, a_h^t), \widetilde{\boldsymbol{w}}_h^t\rangle - \langle\boldsymbol{\phi}(s_h^{t-1}, a_h^{t-1}), \boldsymbol{w}_h^t\rangle + \langle\boldsymbol{\phi}(s_h^{t-1}, a_h^{t-1}), \widetilde{\boldsymbol{w}}_h^t\rangle\right)\right)^2}{\sum_{i=1}^{t-1}\left(\sum_{h=1}^{H}\left(\langle\boldsymbol{\phi}(s_h^i, a_h^i), \boldsymbol{w}_h^t\rangle - \langle\boldsymbol{\phi}(s_h^i, a_h^i), \widetilde{\boldsymbol{w}}_h^t\rangle - \langle\boldsymbol{\phi}(s_h^{i-1}, a_h^{i-1}), \boldsymbol{w}_h^t\rangle + \langle\boldsymbol{\phi}(s_h^{i-1}, a_h^{i-1}), \widetilde{\boldsymbol{w}}_h^t\rangle\right)\right)^2 + \lambda},
\end{aligned}
$$

and

$$\texttt{deed}_T(\mathcal{Q}; \lambda) = \sum_{t=1}^{T} F_t(\mathcal{Q}, \lambda).$$

By definition,

$$
\begin{aligned}
&F_t(\mathcal{Q}, \lambda) \\
&= \frac{\left(\sum_{h=1}^{H}\left(\langle\boldsymbol{\phi}(s_h^t, a_h^t), \boldsymbol{w}_h^t\rangle - \langle\boldsymbol{\phi}(s_h^t, a_h^t), \widetilde{\boldsymbol{w}}_h^t\rangle - \langle\boldsymbol{\phi}(s_h^{t-1}, a_h^{t-1}), \boldsymbol{w}_h^t\rangle + \langle\boldsymbol{\phi}(s_h^{t-1}, a_h^{t-1}), \widetilde{\boldsymbol{w}}_h^t\rangle\right)\right)^2}{\sum_{i=1}^{t-1}\left(\sum_{h=1}^{H}\left(\langle\boldsymbol{\phi}(s_h^i, a_h^i), \boldsymbol{w}_h^t\rangle - \langle\boldsymbol{\phi}(s_h^i, a_h^i), \widetilde{\boldsymbol{w}}_h^t\rangle - \langle\boldsymbol{\phi}(s_h^{i-1}, a_h^{i-1}), \boldsymbol{w}_h^t\rangle + \langle\boldsymbol{\phi}(s_h^{i-1}, a_h^{i-1}), \widetilde{\boldsymbol{w}}_h^t\rangle\right)\right)^2 + \lambda} \\[2mm]
&= \frac{\left(\sum_{h=1}^{H}\langle\boldsymbol{\phi}(s_h^t, a_h^t) - \boldsymbol{\phi}(s_h^{t-1}, a_h^{t-1}), \boldsymbol{w}_h^t - \widetilde{\boldsymbol{w}}_h^t\rangle\right)^2}{\sum_{i=1}^{t-1}\left(\sum_{h=1}^{H}\langle\boldsymbol{\phi}(s_h^i, a_h^i) - \boldsymbol{\phi}(s_h^{i-1}, a_h^{i-1}), \boldsymbol{w}_h^t - \widetilde{\boldsymbol{w}}_h^t\rangle\right)^2 + \lambda} \\[2mm]
&= \frac{\left(\langle\boldsymbol{\phi}_e(\{s_h^t, a_h^t\}_{h=1}^{H}) - \boldsymbol{\phi}_e(\{s_h^{t-1}, a_h^{t-1}\}_{h=1}^{H}), \boldsymbol{w}_{[H]}^t - \widetilde{\boldsymbol{w}}_{[H]}^t\rangle\right)^2}{\sum_{i=1}^{t-1}\left(\langle\boldsymbol{\phi}_e(\{s_h^i, a_h^i\}_{h=1}^{H}) - \boldsymbol{\phi}_e(\{s_h^{i-1}, a_h^{i-1}\}_{h=1}^{H}), \boldsymbol{w}_{[H]}^t - \widetilde{\boldsymbol{w}}_{[H]}^t\rangle\right)^2 + \lambda} \\[2mm]
&= \frac{\left\|\boldsymbol{w}_{[H]}^t - \widetilde{\boldsymbol{w}}_{[H]}^t\right\|^2_{(\boldsymbol{\phi}^t - \boldsymbol{\phi}^{t-1})(\boldsymbol{\phi}^t - \boldsymbol{\phi}^{t-1})^\top}}{\left\|\boldsymbol{w}_{[H]}^t - \widetilde{\boldsymbol{w}}_{[H]}^t\right\|^2_{\sum_{i=1}^{t-1}(\boldsymbol{\phi}^i - \boldsymbol{\phi}^{i-1})(\boldsymbol{\phi}^i - \boldsymbol{\phi}^{i-1})^\top} + \lambda},
\end{aligned}
$$

where

$$\boldsymbol{\phi}^t = \boldsymbol{\phi}_e\left(\{(s_h^t, a_h^t)\}_{h \in [H]}\right), \quad \text{and} \quad \boldsymbol{w}_{[H]}^t = \begin{bmatrix} \boldsymbol{w}_1^t \\ \boldsymbol{w}_2^t \\ \vdots \\ \boldsymbol{w}_H^t \end{bmatrix},$$

are concatenated vectors. Let $\lambda' = \lambda / \left\|\boldsymbol{w}_{[H]}^t - \widetilde{\boldsymbol{w}}_{[H]}^t\right\|_I^2$, we have that

$$
\begin{aligned}
&\lambda + \left\|\boldsymbol{w}_{[H]}^t - \widetilde{\boldsymbol{w}}_{[H]}^t\right\|^2_{\sum_{i=1}^{t-1}(\boldsymbol{\phi}^i - \boldsymbol{\phi}^{i-1})(\boldsymbol{\phi}^i - \boldsymbol{\phi}^{i-1})^\top} \\
&= \left\|\boldsymbol{w}_{[H]}^t - \widetilde{\boldsymbol{w}}_{[H]}^t\right\|^2_{\sum_{i=1}^{t-1}(\boldsymbol{\phi}^i - \boldsymbol{\phi}^{i-1})(\boldsymbol{\phi}^i - \boldsymbol{\phi}^{i-1})^\top + \lambda' I}.
\end{aligned}
$$

As a result,

$$\frac{\left(\sum_{h=1}^{H}\langle\phi(s_h^t,a_h^t),\boldsymbol{w}_h^t\rangle - \langle\phi(s_h^t,a_h^t),\widetilde{\boldsymbol{w}}_h^t\rangle - \langle\phi(s_h^{t-1},a_h^{t-1}),\boldsymbol{w}_h^t\rangle + \langle\phi(s_h^{t-1},a_h^{t-1}),\widetilde{\boldsymbol{w}}_h^t\rangle\right)^2}{\sum_{i=1}^{t-1}\left(\sum_{h=1}^{H}\langle\phi(s_h^i,a_h^i),\boldsymbol{w}_h^t\rangle - \langle\phi(s_h^i,a_h^i),\widetilde{\boldsymbol{w}}_h^t\rangle - \langle\phi(s_h^{i-1},a_h^{i-1}),\boldsymbol{w}_h^t\rangle + \langle\phi(s_h^{i-1},a_h^{i-1}),\widetilde{\boldsymbol{w}}_h^t\rangle\right)^2 + \lambda}$$

$$=\frac{\left\|\boldsymbol{w}_{[H]}^t - \widetilde{\boldsymbol{w}}_{[H]}^t\right\|^2_{\left(\phi^t-\phi^{t-1}\right)\left(\phi^t-\phi^{t-1}\right)^\top}}{\left\|\boldsymbol{w}_{[H]}^t - \widetilde{\boldsymbol{w}}_{[H]}^t\right\|^2_{\sum_{i=1}^{t-1}\left(\phi^i-\phi^{i-1}\right)\left(\phi^i-\phi^{i-1}\right)^\top + \lambda'I}}$$

$$\leq \max_{\boldsymbol{w}|\boldsymbol{w}\leq 4H\sqrt{Hd}} \frac{\left\|\boldsymbol{w}\right\|^2_{\left(\phi^t-\phi^{t-1}\right)\left(\phi^t-\phi^{t-1}\right)^\top}}{\left\|\boldsymbol{w}\right\|^2_{\sum_{i=1}^{t-1}\left(\phi^i-\phi^{i-1}\right)\left(\phi^i-\phi^{i-1}\right)^\top + \lambda'I}}.$$

According to the generalized Rayleigh-Ritz quotient method (Croot (2005), you can also refer to the section 4.3 in Ghojogh et al. (2023)), the solution of the problem

$$\max_{\boldsymbol{w}|\boldsymbol{w}\leq 4H\sqrt{Hd}} \frac{\left\|\boldsymbol{w}\right\|^2_{\left(\phi^t-\phi^{t-1}\right)\left(\phi^t-\phi^{t-1}\right)^\top}}{\left\|\boldsymbol{w}\right\|^2_{\sum_{i=1}^{t-1}\left(\phi^i-\phi^{i-1}\right)\left(\phi^i-\phi^{i-1}\right)^\top + \lambda'I}}$$

is the maximum eigenvalue of

$$M_t = \left(\sum_{i=1}^{t-1}\left(\phi^i - \phi^{i-1}\right)\left(\phi^i - \phi^{i-1}\right)^\top + \lambda'I\right)^{-1}\left(\phi^t - \phi^{t-1}\right)\left(\phi^t - \phi^{t-1}\right)^\top.$$

Since for any matrix $B$ and non-zero vector $\phi$, $\text{rank}(B^{-1}\phi\phi^\top) \leq \text{rank}(\phi\phi^\top) = 1$, $M_t$ only has one linearly independent eigenvector, which is

$$v = \left(\sum_{i=1}^{t-1}\left(\phi^i - \phi^{i-1}\right)\left(\phi^i - \phi^{i-1}\right)^\top + \lambda'I\right)^{-1}\left(\phi^t - \phi^{t-1}\right).$$

We can multiply them together,

$$M_t\left(\sum_{i=1}^{t-1}\left(\phi^i - \phi^{i-1}\right)\left(\phi^i - \phi^{i-1}\right)^\top + \lambda'I\right)^{-1}\left(\phi^t - \phi^{t-1}\right)$$

$$=\underbrace{\left(\sum_{i=1}^{t-1}\left(\phi^i - \phi^{i-1}\right)\left(\phi^i - \phi^{i-1}\right)^\top + \lambda'I\right)^{-1}\left(\phi^t - \phi^{t-1}\right)}_{\text{the eigenvector}} \cdot \underbrace{\left(\phi^t - \phi^{t-1}\right)^\top\left(\sum_{i=1}^{t-1}\left(\phi^i - \phi^{i-1}\right)\left(\phi^i - \phi^{i-1}\right)^\top + \lambda'I\right)^{-1}\left(\phi^t - \phi^{t-1}\right)}_{\text{the eigenvalue}}$$

$$=\left(\left(\phi^t - \phi^{t-1}\right)^\top\left(\sum_{i=1}^{t-1}\left(\phi^i - \phi^{i-1}\right)\left(\phi^i - \phi^{i-1}\right)^\top + \lambda'I\right)^{-1}\left(\phi^t - \phi^{t-1}\right)\right)v.$$

So the solution is

$$\left(\phi^t - \phi^{t-1}\right)^\top\left(\sum_{i=1}^{t-1}\left(\phi^i - \phi^{i-1}\right)\left(\phi^i - \phi^{i-1}\right)^\top + \lambda'I\right)^{-1}\left(\phi^t - \phi^{t-1}\right).$$

Let $\Sigma_{t-1} = \sum_{i=1}^{t-1} (\phi^i - \phi^{i-1})(\phi^i - \phi^{i-1})^\top + \lambda' I$, as a result of the above argument,

$$\texttt{deed}_T(\mathcal{Q}; \lambda) = \sum_{t=1}^{T} F_t(\mathcal{Q}, \lambda)$$

$$\leq \sum_{t=1}^{T} (\phi^t - \phi^{t-1})^\top \Sigma_{t-1}^{-1} (\phi^t - \phi^{t-1}).$$

By the elliptical potential lemma in the following:

**Lemma B.7** (Elliptical Potential Lemma; For Linear Updating). *Let $\{\phi_t\}_{t=1}^{\infty}$ be a sequence in $\mathbb{R}^d$, $\mathbf{U}$ a $d \times d$ positive definite matrix and define $\mathbf{U}_t = \mathbf{U} + \sum_{i=1}^{t} \phi_i^\top \phi_i$. If $\|\phi_i\|_2 \leq L$ and $\lambda_{\min}(\mathbf{U}) \geq \max(1, L^2)$, then we have*

$$\sum_{i=1}^{t} \phi_i^\top (\mathbf{U}_{i-1})^{-1} \phi_i \leq 2 \log \left( \frac{\det(\mathbf{U}_t)}{\det(\mathbf{U})} \right).$$

We have

$$\sum_{t=1}^{T} (\phi^t - \phi^{t-1})^\top \Sigma_{t-1}(\phi^t - \phi^{t-1}) \leq 2 \log \left( \frac{\det(\Sigma_t)}{\det(\lambda' I)} \right).$$

Let $\{\nu_j\}_{j=1}^{dH}$ be the eigenvalues of $\Sigma_t$, and $A = \sum_{t=1}^{T}(\phi^t - \phi^{t-1})(\phi^t - \phi^{t-1})^\top$, then by AM-GM,

$$\det(\Sigma_t) = \prod_{j=1}^{dH} \nu_j \leq \left( \frac{\sum_{j=1}^{dH} \nu_j}{dH} \right)^{dH} = \left( \frac{\text{tr}(\Sigma_t)}{dH} \right)^{dH} = \left( \lambda' + \frac{\text{tr}(A)}{dH} \right)^{dH} \leq \left( \lambda' + \frac{4TH}{dH} \right)^{dH} \leq \left( \lambda' + \frac{4T}{d} \right)^{dH}.$$

Also by the definition, $\det(\lambda' I) = (\lambda')^{dH}$. As a result,

$$\sum_{t=1}^{T} (\phi^t - \phi^{t-1})^\top \Sigma_{t-1}(\phi^t - \phi^{t-1}) \leq 2 \log \left( \frac{\det(\Sigma_t)}{\det(\lambda' I)} \right)$$

$$\leq 2dH \log \left( \frac{\lambda' + 4T/d}{\lambda'} \right).$$

Note that

$$\lambda' = \lambda / \left\| \boldsymbol{w}_{[H]}^t - \widetilde{\boldsymbol{w}}_{[H]}^t \right\|_I^2$$

$$= \lambda / \left\| \boldsymbol{w}_{[H]}^t - \widetilde{\boldsymbol{w}}_{[H]}^t \right\|_2^2 \geq \frac{\lambda}{16dH^3}.$$

This means

$$\frac{\lambda' + 4T/d}{\lambda'} \leq \frac{\lambda + 64TH^3}{\lambda}.$$

Finally,

$$\texttt{deed}_T(\mathcal{Q}; \lambda) \leq 2dH \log \left( \frac{\lambda + 64TH^3}{\lambda} \right).$$

This completes the proof. □

