# OpenReview forum: "Rethinking the Hardness of PbRL: A Provable General Regret Bound"
_ICML.cc/2026/Conference — ICML 2026 regular_

### Official Review · Reviewer_rfmb · 2026-03-07

**Soundness:** 3
**Presentation:** 4
**Significance:** 3
**Originality:** 4
**Overall Recommendation:** 5
**Confidence:** 4

**Summary:**

This paper revisits the question of how to characterize the hardness of preference-based reinforcement learning with general value function approximation.
The paper introduces DEED (Dual Episodic Eluder Dimension), a complexity measure that separates the uncertainty arising from transition dynamics and reward functions. A key technical step reformulates the learning problem so that trajectory-level comparisons can be analyzed through per-step reward structure, which enables the regret analysis under the DEED framework.
The paper derives regret guarantees expressed in terms of DEED and discusses how these bounds relate to existing results, including the Linear MDP setting.

**Compliance With Llm Reviewing Policy:**

Affirmed.

**Final Justification:**

I have reviewed the authors’ response, and my concerns have been fully addressed. I will therefore maintain my current positive score.

**Key Questions For Authors:**

1. In Line 779, the notation $\hat{p} \sim p$ appears, but it is unclear whether $p$ refers to $p_R$ or $p_{R^\*}$. From the context, it seems that $p_{R^\*}$ may be intended. Could the authors clarify this point?

2. In Line 1643, there appears to be an issue with the reference citation.

3. The regret bound introduces an additional factor of $H$. Is this due to an inherent limitation arising from comparing quantities at the *trajectory level*, or do the authors believe that this factor could potentially be removed with a tighter technical analysis?

**Limitations:**

Yes.

The paper explicitly discusses its limitations and outlines potential future directions, particularly regarding the tightness of the regret bound and possible refinements of the DEED complexity measure. However, as mentioned in the weaknesses section, it would also be useful to comment on the practical tractability of the set-constrained optimization appearing in the pseudocode, as this may present challenges in real implementations.

**Strengths And Weaknesses:**

## Strengths

### 1. Clarity and presentation

- The paper is very clearly written and structurally well organized. In particular, the exposition is accessible even for readers who are not deeply familiar with general value function approximation. The authors carefully introduce terminology and provide detailed explanations of the background literature, which makes the paper easier to follow despite the technical nature of the topic.

- Another positive aspect is that prior work is accurately acknowledged and the contributions of the paper are clearly positioned relative to existing results. The novelty of the proposed framework and the role of DEED are explicitly stated, which helps clarify the contribution of the paper.

- I also appreciated the pacing of the theoretical development. The transitions between major steps in the analysis are well motivated: the authors often point out the limitations of existing analytical approaches and then introduce the next idea as a natural workaround. This structure makes the progression of the theory easy to follow and improves the overall readability of the paper.

### 2. Technical insight

- I found the transformation introduced in Equation (6) particularly interesting. The reformulation that enables trajectory-level analysis through per-step rewards provides an appealing analytical perspective and appears to be a novel technical component of the work. This formulation helps bridge the structure of the reward signal with the complexity measure used in the regret analysis.

- Regarding the technical correctness of the paper, I read through the proofs including those in the appendix. For arguments that closely follow existing results, I mainly checked that the derivations were consistent with the prior literature. For the parts that correspond to the authors’ novel setup, I examined the derivations more carefully. Based on my reading, I did not find any obvious critical errors in the proofs.

## Weaknesses

- The main weakness of the paper is the absence of empirical validation. However, since the contribution is primarily theoretical, I do not consider this to be a major drawback. That said, it would be helpful if the authors discussed potential difficulties that may arise in practical implementations. In particular, the set-constrained optimization appearing in the pseudocode may not always be tractable in practice. A short discussion of such practical challenges in the limitations section could be valuable for future researchers who attempt to build on this framework.

---

> ### Author Rebuttal · Authors · 2026-03-30
>
> We thank the reviewer for their positive assessment of our work, particularly regarding the clarity of our presentation and the technical novelty of the DEED framework. We appreciate the constructive suggestions regarding practical tractability and the identification of minor notation issues.
>
> **Q1: In Line 779, the notation $\hat{p}\sim p$ appears, but it is unclear whether $p$ refers to $p_{R^\star}$ or $p_R$. From the context, it seems that $p_{R^\star}$ may be intended. Could the authors clarify this point?**
>
> The reviewer is correct. The notation $p$ in that context refers to the ground-truth preference distribution $p_{R^\star}$. In our revision, we will explicitly denote this as $p_{R^\star}$ to avoid any ambiguity, clarifying that $\hat{p}$ represents the empirical preferences sampled from the true underlying reward structure.
>
> **Q2: In Line 1643, there appears to be an issue with the reference citation.**
>
> Thank you for pointing out this issue. We apologize for the missing citation. This refers to:
> > *Vershynin, R. (2011). Introduction to the non-asymptotic analysis of random matrices. arXiv preprint arXiv:1011.3027.*
>
> We have updated our bibliography and will ensure all references are correctly linked in the final version.
>
> **Q3: The regret bound introduces an additional factor of $H$. Is this due to an inherent limitation arising from comparing quantities at the trajectory level, or do the authors believe that this factor could potentially be removed with a tighter technical analysis?**
>
> We believe that this factor of $H$ is an inherent limitation arising from comparing quantities at trajectory level. The reviewer can image in the outcome based RL, which is strictly easier than PbRL, a small twist of the outcome feedback can be hide in the long horizon.
>
> We believe this factor of $H$ represents an **inherent limitation** stemming from comparing feedback at the trajectory level rather than the step level.
>
> To provide intuition, consider standard outcome-based Reinforcement Learning (RL), which is a strictly simpler setting than Preference-based RL (PbRL). Even in that setting, a subtle shift in outcome feedback can be "diluted" or obscured over a long horizon $H$. Because PbRL relies on preference signals over entire trajectories rather than immediate step-wise rewards, the uncertainty compounds, making this factor a structural hurdle rather than a loose end in the technical analysis.
>
> **W1: The main weakness of the paper is the absence of empirical validation. However, since the contribution is primarily theoretical, I do not consider this to be a major drawback. That said, it would be helpful if the authors discussed potential difficulties that may arise in practical implementations.**
>
> We thank the reviewer for this suggestion and will revise accordingly. Below we provide a brief discussion of computational tractability.
>
> In practice, the function approximation steps may require training highly expressive models (e.g., deep neural networks), leading to non-convex and computationally demanding optimization problems. As such, analyzing their algorithmic tractability is beyond the scope of this work. Our results should therefore be interpreted as providing *statistical guarantees*, rather than worst-case computational efficiency.
>
> However, in the special case of *linear function approximation*, there exists a well-established body of work on computationally efficient optimization methods (Tsitsiklis and Van Roy 1997, Jaggi 2013, Jin et al., 2019, Mhammedi 2024). In such settings, these techniques can be incorporated as plug-in components to yield tractable implementations.
>
> ### References
>
> Jin, C., Yang, Z., Wang, Z., and Jordan, M. I. Provably efficient reinforcement learning with linear function approximation, 2019. URL https://arxiv.org/abs/1907.05388.
>
> Jaggi, M. Revisiting Frank-Wolfe: Projection-free sparse convex optimization. Proceedings of the 30th International Conference on Machine Learning, PMLR 28(1):427-435, 2013. URL https://proceedings.mlr.press/v28/jaggi13.html.
>
> Mhammedi, Z. Sample and oracle efficient reinforcement learning for MDPs with linearly-realizable value functions, 2024. URL https://arxiv.org/abs/2409.04840. arXiv:2409.04840.
>
> Tsitsiklis, J. N. and Van Roy, B. An analysis of temporal-difference learning with function approximation, 1997. URL https://www.mit.edu/~jnt/Papers/J063-97-bvr-td.pdf.

---

> > ### Author Rebuttal · Reviewer_rfmb · 2026-04-03
> >
> > Thank you for the detailed and thoughtful responses. My concerns have been sufficiently addressed through the rebuttal, and I encourage the authors to incorporate these clarifications into the final revision. Based on this, I am inclined to maintain a positive score.

---

### Official Review · Reviewer_FWjg · 2026-03-10

**Soundness:** 3
**Presentation:** 3
**Significance:** 3
**Originality:** 2
**Overall Recommendation:** 4
**Confidence:** 3

**Summary:**

This paper studies preference-based reinforcement learning (PbRL) with trajectory-level comparison feedback instead of scalar rewards. It proposes Recursive Trajectory-based Preference Q-Learning (RTPQ) and proves a general $\tilde O(\sqrt{T})$ regret bound under function approximation. The main technical idea is a new complexity measure, Dual Episodic Eluder Dimension (DEED), designed to capture both trajectory-level and comparison-based aspects of PbRL. For linear MDPs, the paper derives a bound of $\tilde O(dH\sqrt{T}\max\{H^{3/2},1/\kappa\})$. Overall, this submission investigates the key problem of whether PbRL can admit general, provable regret guarantees beyond restrictive special cases.

**Compliance With Llm Reviewing Policy:**

Affirmed.

**Final Justification:**

The rebuttal addressed my concerns on the seemingly strong assumptions and strengthen my confidence on the positive evaluation.

**Key Questions For Authors:**

1. Can you give a more intuitive explanation of DEED and why standard eluder-style notions are insufficient here?  A clear answer would increase confidence in the conceptual novelty.
2. What is the computational complexity of RTPQ in practice?
   If it is efficient beyond theory, my assessment would improve.
3. How robust are the guarantees to preference-model misspecification?
   A satisfactory answer would strengthen the soundness and relevance of the result.

**Limitations:**

yes

**Strengths And Weaknesses:**

**Strength**
- Timely theoretical problem, especially given the relevance of PbRL/RLHF.
- The DEED notion is interesting and appears designed to the distinctive structure of preference feedback.
- The analysis aims to unify PbRL with special cases such as dueling bandits and outcome-based RL.
- Overall, a notable aspect discussed by the manuscript is its explicit treatment of both comparison-based and outcome-based structure in a single regret framework.

**Weaknesses**
- The paper is quite dense, and the intuition behind DEED and the regret decomposition could be clearer.
- The contribution seems mainly analytical; the algorithmic novelty itself feels more modest.
- The assumptions may be strong, and the practical implications are unclear without experiments.

---

> ### Author Rebuttal · Authors · 2026-03-30
>
> We thank the reviewer for the positive feedback on our work's timeliness and the novelty of DEED. We address the questions and weaknesses below and will incorporate these clarifications in the revision.
>
> ### Q1/W1: Intuition & DEED vs. standard eluder dimension
> Standard eluder-style notions are designed for additive, step-wise rewards. They face two primary hurdles in PbRL:
>
> 1.  **Feedback Granularity:** Standard notions assume $\ell_2$ errors can be decomposed per step. In PbRL, we observe comparison for trajectories. Controlling step-wise error with trajectory-level feedback is ill-posed for standard Eluder dimensions unless one focuses model-based functions, which adds significant complexity.
> 2.  **Relative Signals:** In contrast to the reward/return, a policy's quality from preference is relative. An increase in the return of one policy may be masked by an increase in the baseline it is compared against. DEED explicitly incorporates pairs of trajectories into the complexity measure.
>
> **Intuition:** Much like the standard Eluder dimension, DEED quantifies the discrepancy between 'known' and 'unknown' of the function space, which are represented by the denominator and numerator within DEED. However, DEED is specifically *lifted to the trajectory pairs*. By measuring this discrepancy over pairs, it directly captures the degree to which new preferences reduce our uncertainty about which policy is truly preferred.
>
> ### Q2: Computational complexity of RTPQ
> Our focus is on the *information-theoretic and statistical limits* of PbRL under general function approximation.
>
> In practical applications, these approximation steps involve training highly expressive models, such as deep neural networks. Because these optimization landscapes are inherently non-convex and computationally intensive, their algorithmic tractability is outside the immediate scope of this paper. Consequently, our results should be interpreted as *statistical guarantees* rather than claims of worst-case computational efficiency.
>
> However, in the specific regime of *linear function approximation*, there is a robust body of literature surrounding computationally tractable methods [5, 6, 7, 8]. For these cases, well-established optimization techniques can be integrated as a "plug-in" component to ensure efficient implementation.
>
> ### Q3: Misspecification Robustness
> Our results are determined by the link function’s properties, reflecting the sensitivity of preference to performance variations.
> * **Structural Misspecification:** If the preference deviates from the assumed link function, the regret bound scales gracefully according to the change in these curvature constants ($\kappa$).
> * **Adversarial/Noisy Feedback:** While cases where the preference is diametrically opposed to the true performance are beyond our current scope, the algorithm's performance degrades proportionally to the degree and location of the misspecification.
>
> ### W2: Algorithmic/Analytical Novelty
> The novelty of our work is twofold, encompassing both algorithmic design and theoretical framework:
> * **Algorithmic Synergy:** As detailed in the Introduction, our algorithm integrates five designs—ranging from trajectory-wise estimation to consecutive comparisons. While individual elements may have appeared in disparate contexts, their specific synthesis into a cohesive framework is a distinct contribution.
> * **Analytical Novelty:** A core innovation of our work is the analytical bridge provided by DEED. It allows us to unify preference signals with value-function approximation, a gap that previous literature [2, 3, 4] has struggled to close without restrictive assumptions.
>
> ### W3: Assumptions/Practicality
> Below, we clarify the roles and practical relevance of our assumptions:
> * **A. 2.2 (Link Function):** This is intended as a unified condition that encompasses widely used preference models. It measures how sensitive preference changes are relative to performance gains; it is a problem-dependent constant that stays bounded in practical scenarios.
> * **A. 2.4 (Realizability):** This is a standard requirement in function approximation [2, 3, 4]. In modern applications, we employ high-capacity models where the approximation error can be driven toward zero.
> * **A. 2.6 (Completeness):** Completeness is a foundational requirement in value function estimation [1]. Notably, existing lower bounds [1] demonstrate that, without this assumption, many RL algorithms risk divergence, making it a critical pillar for our analysis.
>
> ---
> **References**
> [1] Chen & Jiang (2019); [2] Chen et al. (2022); [3] Wang et al. (2023); [4] Chen et al. (2025a); [5] Jin et al. (2019)
>
> [6] Tsitsiklis, J. N. and Van Roy, B. An analysis of temporal-difference learning with function approximation, 1997
>
> [7] Jaggi, M. Revisiting Frank-Wolfe: Projection-free sparse convex optimization. 2013
>
> [8] Mhammedi, Z. Sample and oracle efficient reinforcement learning for MDPs with linearly-realizable value functions, 2024

---

> > ### Author Rebuttal · Reviewer_FWjg · 2026-04-02
> >
> > Thank you very much for the clarification. I will keep my positive evaluation.

---

### Official Review · Reviewer_1HgT · 2026-03-13

**Soundness:** 2
**Presentation:** 3
**Significance:** 3
**Originality:** 3
**Overall Recommendation:** 4
**Confidence:** 3

**Summary:**

The paper studies preference-based reinforcement learning (PbRL) with trajectory-level, comparison-based feedback under general function approximation. It proposes a value-based algorithm, Recursive Trajectory-based Preference Q-learning (RTPQ), that estimates a trajectory-return model and a backup function for value learning and optimistic selection, and claims a general $\sqrt{T}$ regret bound in terms of a new complexity measure, the Dual Episodic Eluder Dimension (DEED). The paper also claims a linear-MDP specialization, where DEED scales as $O(dH)$ and the regret bound becomes $\tilde O(dH \sqrt{T} \max\{H^{3/2}, 1 / \kappa\})$, and claims recoveries of known results in special cases such as dueling bandits and standard outcome-based RL.

**Compliance With Llm Reviewing Policy:**

Affirmed.

**Final Justification:**

The rebuttal addressed my main concerns. I changed my overall evaluation from 3 to 4.

**Key Questions For Authors:**

1. Can the authors clarify the exact dependence of $\kappa$ on the return range for Bradley-Terry or logistic links, and restate the assumption accordingly?
2. What optimization oracles or tractability assumptions are needed for the ERM steps and the optimistic selection?
3. Can the paper provide formal corollaries or a more precise comparison showing how the dueling-bandit and outcome-based RL cases are recovered, and how DEED differs from generalized-eluder notions?
4. Can the authors clarify the centering/sign convention in Eq. (15)-(20), Lemma A.9, and the T1 step of Theorem A.11, and explain how this matches the DEED definition in Definition 2.10 and the Bellman equation for $Q^*$?

I am not fully familiar with this line of work, and I would be open to changing my assessment after reading the authors' clarification.

**Limitations:**

The paper already mentions passive querying and the linear-MDP-only instantiation, but it should also discuss the strength of Assumptions 2.2/2.4/2.6 and make the computational assumptions behind its optimization steps more explicit.

**Strengths And Weaknesses:**

Strengths

- The paper proposes a value-based algorithm for PbRL with trajectory-level preference feedback and introduces the DEED complexity measure.
- The dual regret decomposition and backup-estimation mechanism are technically interesting, and the linear-MDP specialization could be meaningful if the proof issues are repaired.
- The paper aims to connect PbRL, outcome-based RL, and dueling-bandit settings through a common complexity measure.

Weaknesses

- The paper makes strong assumptions, especially around the link-function condition and the role of $\kappa$.
- The paper does not make its computational assumptions explicit: the optimization steps over the function classes and the optimistic selection appear oracle-based, but the paper does not specify what tractability or oracle guarantees are assumed.
- I had difficulty understanding the sign/centering in Eq. (15)-(20), Lemma A.9, and the T1/DEED part of Theorem A.11, and I would appreciate clarification on whether the current derivation is correct as written.
- The paper claims recoveries of dueling bandits and outcome-based RL, and introduces DEED as related to generalized-eluder notions, but these connections are not made explicit through formal corollaries or a careful comparison.

---

> ### Author Rebuttal · Authors · 2026-03-30
>
> We thank the reviewer for the thoughtful feedback and constructive questions. We address each concern below and will incorporate these clarifications in the revision.
>
> ### Q1/W1: Link function
> Our link-function assumption (A 2.2) is intended as a *unified condition* that captures commonly used models such as Bradley-Terry and logistic links. The parameter $\kappa$ quantifies the **inverse slope (curvature)** of the link function over the relevant range of return differences.
>
> In standard settings (e.g., [1, 2, 3, 4, 5]), it is natural to assume that returns are bounded in an interval $[0, R]$. This implies that pairwise return differences lie in $[-R, R]$. For logistic/Bradley-Terry models, the derivative of the link function is uniformly bounded on this interval. Consequently, both $\kappa$ and $\kappa'$ can be taken as finite constants depending only on $R$.
>
> ### Q2/W2: Computational assumptions and oracle access
> We thank the reviewer for pointing this out. Our paper focuses on the information-theoretic and statistical limits of PbRL under function approximation. Making explicit computational assumptions is not the goal here; characterizing statistical limits is typically the first step before addressing computational efficiency.
>
> While our algorithm uses ERM-style optimization and optimistic selection steps, assuming implicit access to an ERM oracle is standard in theoretical RL [3, 4, 5, 6]. In practice, these ERM steps correspond to training highly expressive models, such as neural networks, which are non-convex and computationally intensive. As such, their tractability is not the focus of this work, and our results should be interpreted primarily as statistical guarantees rather than statements about efficient computation.
>
> However, in the specific regime of *linear function approximation*, there is a robust body of literature surrounding computationally tractable methods [6, 8, 9, 10]. For these cases, well-established optimization techniques can be integrated as a "plug-in" component to ensure efficient implementation.
>
> ### Q3/W4: Connections to dueling bandits and outcome-based RL
> We appreciate the opportunity to clarify these points:
>
> **Dueling bandits.**
> This setting is a special case of linear MDPs with horizon ($H = 1$). Applying Proposition 4.2 with ($H = 1$) directly yields a regret bound of $\tilde{\mathcal{O}}(d \sqrt{T})$, which matches existing results such as [11].
>
> **Outcome-based RL.**
> PbRL is structurally harder than standard outcome-based RL (preference signals can be generated using the returns). By simply taking the returns from outcome-based rl, sampling the preferences, and feeding them into RTPQ, we recover the upper bound presented in Theorem 4.1 for outcome-based RL.
>
> **Relation to eluder dimension.**
> DEED differs from standard generalized eluder dimension by incorporating *pairs of trajectories* into the complexity measure, reflecting the two defining aspects of PbRL: trajectory-level feedback and comparison-based signals.
>
> ### Q4/W3: Sign/centering issues
> The key point is that the quantities in Eq. (15-20), Def. 2.10 (DEED), and Lemma A.9 are all expressed in terms of *squared loss*.
> As a result, *sign and centering do not affect the bounds*, since the analysis depends only on squared deviations.
>
> In the proof of Theorem A.11 (including $T_1$), the relevant expressions (as what we do for the Bellman-style recursion) are controlled via these $\ell_2$-type quantities in the latter part of the proof, which ensures the consistency with the DEED and the quantities mentioned before.
>
> ### **Further discussing/ limitations**
> We appreciate the suggestion to better contextualize our assumptions:
>
> * **Assumption 2.2 (link function)**: As discussed above, this is a natural generalization covering standard models.
> * **A 2.4 (realizability)**: This is standard in function approximation and reflects the expressive power of the hypothesis class.
> * **A 2.6 (completeness)**: Assumption 2.6 is standard in value function estimation [7]. Although its practical implications remain under-studied, it provides the theoretical foundation for many reinforcement learning algorithms used in real-world applications.
> * **Computational Assumptions:** As discussed before, we focus on the statistical perspect of PbRL. Explicit computational assumptions are beyond this paper’s scope.
>
> ---
> **References**
> [1] Novoseller et al. (2020); [2] Xu et al. (2020); [3] Chen et al. (2022); [4] Wang et al. (2023); [5] Chen et al. (2025a); [6] Jin et al. (2019); [7] Chen & Jiang (2019).
>
> [8] Tsitsiklis, J. N. and Van Roy, B. An analysis of temporal-difference learning with function approximation, 1997
>
> [9] Jaggi, M. Revisiting Frank-Wolfe: Projection-free sparse convex optimization. 2013
>
> [10] Mhammedi, Z. Sample and oracle efficient reinforcement learning for MDPs with linearly-realizable value functions, 2024
>
> [11] Xuheng Li, Heyang Zhao, and Quanquan Gu. Feel-good thompson sampling for contextual dueling bandits, 2024

---

> > ### Author Rebuttal · Reviewer_1HgT · 2026-04-02
> >
> > Thank you for the rebuttal. The clarifications on the statistical/oracle framing are helpful, and I am keeping my current score for now.
> >
> > My main remaining concern is the sign/centering consistency between Eq. (15)-(20), Definition 2.10 (DEED), Lemma A.9, and the T1 step in Theorem A.11. The issue is whether the expression inside the square matches the Bellman-residual / DEED form. Concretely, when I plug in $Q = Q^\star$ and use the Bellman equation $Q^\star = r^\star + P^{\pi^\star} Q^\star$, Eq. (15)--(17) seems to give another copy of the return-difference term rather than cancellation; and in Theorem A.11, T1 is written using $D(Q_t, P^{\pi_t} Q_t - r; \cdot, \cdot)$, whereas Definition 2.10 is stated with $Q' \in r^\star + P^\star Q$.

---

> > > ### Author Response · Authors · 2026-04-02
> > >
> > > We sincerely thank the reviewer for their careful reading and for engaging with our rebuttal. We appreciate you highlighting this concern, and we would like to clarify the centering in Eq. (15)-(17) and acknowledge a typo in the appendix that caused the inconsistency regarding $T_1$.
> > >
> > > **Regarding Eq. (15)-(17):**
> > > As indicated in Eq. (9), our operator $\mathcal{B}Q^\star$ is explicitly designed to approximate $P^{\pi^\star} Q^\star$ alone, rather than the standard full Bellman update $r^\star + P^{\pi^\star} Q^\star$. The "missing" reward term $r$ is directly accounted for by the empirical return $R$ in Eq. (17). Therefore, when plugging in $Q = Q^\star$, the terms *do* correctly cancel out once $R$ is factored into the equation. This deliberate decoupling of the return $R$ and $P^{\pi^\star} Q^\star$ reflects the nature of Preference-based RL (PbRL), where return and trajectory information are collected as disjoint signals.
> > >
> > > **Regarding $T_1$ in Theorem A.11 and Definition 2.10:**
> > > You are completely correct to flag the sign inconsistency here—this is indeed a typo in our proof between lines 1450 and 1463. As established in line 1444, the formula is $Q^t_h - P^{\pi^t}Q^t - r$, which algebraically groups to $Q^t_h - (P^{\pi^t}Q^t + r)$. Consequently, the DEED term invoked in line 1451 and onwards should be written as $D(Q_t, P^{\pi_t} Q_t + r; \cdot, \cdot)$. With this sign corrected, the expression inside the square perfectly matches the $Q' \in r^\star + P^\star Q$ form given in Definition 2.10.
> > >
> > > We are extremely grateful you caught this. We will carefully correct the signs in the $T_1$ steps in the revised appendix to ensure strict consistency with the DEED definition.

---

### Decision · Program_Chairs · 2026-04-30

**Decision:**

Accept (regular)

**Comment:**

After reading the reviews, the rebuttal, and the discussion with the reviewers, I believe that the paper meets the bar for acceptance at ICML. The main reasons are:

- The paper provides a technically strong and timely theoretical contribution to preference-based reinforcement learning, introducing a novel complexity measure and regret analysis under general function approximation.
- The reviewers found the analysis meaningful and the presentation clear, with the paper offering useful connections to related settings such as dueling bandits and outcome-based RL.
- The rebuttal successfully addressed the main concerns, including clarifications of the assumptions, computational framing, and proof details, which increased confidence in the correctness and significance of the work.

If accepted, the authors are encouraged to incorporate the reviewers’ suggestions into the camera-ready version.